# Learning Mixtures of Ranking Models*

**Pranjal Awasthi**
Princeton University
pawashti@cs.princeton.edu

**Avrim Blum**
Carnegie Mellon University
avrim@cs.cmu.edu

**Or Sheffet**
Harvard University
osheffet@seas.harvard.edu

**Aravindan Vijayaraghavan**
New York University
vijayara@cims.nyu.edu

## Abstract

This work concerns learning probabilistic models for ranking data in a heterogeneous population. The specific problem we study is learning the parameters of a *Mallows Mixture Model*. Despite being widely studied, current heuristics for this problem do not have theoretical guarantees and can get stuck in bad local optima. We present the first polynomial time algorithm which provably learns the parameters of a mixture of two Mallows models. A key component of our algorithm is a novel use of tensor decomposition techniques to learn the top-$k$ prefix in both the rankings. Before this work, even the question of *identifiability* in the case of a mixture of two Mallows models was unresolved.

## 1 Introduction

Probabilistic modeling of ranking data is an extensively studied problem with a rich body of past work [1, 2, 3, 4, 5, 6, 7, 8, 9]. Ranking using such models has applications in a variety of areas ranging from understanding user preferences in electoral systems and social choice theory, to more modern learning tasks in online web search, crowd-sourcing and recommendation systems. Traditionally, models for generating ranking data consider a homogeneous group of users with a *central ranking* (permutation) $\pi^*$ over a set of $n$ elements or alternatives. (For instance, $\pi^*$ might correspond to a "ground-truth ranking" over a set of movies.) Each individual user generates her own ranking as a noisy version of this one central ranking and independently from other users. The most popular ranking model of choice is the *Mallows model* [1], where in addition to $\pi^*$ there is also a scaling parameter $\phi \in (0, 1)$. Each user picks her ranking $\pi$ w.p. proportional to $\phi^{d_{\mathsf{kt}}(\pi, \pi^*)}$ where $d_{\mathsf{kt}}(\cdot)$ denotes the Kendall-Tau distance between permutations (see Section 2).[1] We denote such a model as $\mathcal{M}_n(\phi, \pi^*)$.

The Mallows model and its generalizations have received much attention from the statistics, political science and machine learning communities, relating this probabilistic model to the long-studied work about voting and social choice [10, 11]. From a machine learning perspective, the problem is to find the parameters of the model — the central permutation $\pi^*$ and the scaling parameter $\phi$, using independent samples from the distribution. There is a large body of work [4, 6, 5, 7, 12] providing efficient algorithms for learning the parameters of a Mallows model.

In many scenarios, however, the population is heterogeneous with multiple groups of people, each with their own central ranking [2]. For instance, when ranking movies, the population may be divided into two groups corresponding to men and women; with men ranking movies with one underlying central permutation, and women ranking movies with another underlying central permutation. This naturally motivates the problem of learning a *mixture* of multiple Mallows models for rankings, a problem that has received significant attention [8, 13, 3, 4]. Heuristics like the EM algorithm have been applied to learn the model parameters of a mixture of Mallows models [8]. The problem has also been studied under distributional assumptions over the parameters, e.g. weights derived from a Dirichlet distribution [13]. However, unlike the case of a single Mallows model, algorithms with provable guarantees have remained elusive for this problem.

In this work we give the *first polynomial time algorithm* that *provably* learns a mixture of two Mallows models. The input to our algorithm consists of i.i.d random rankings (samples), with each ranking drawn with probability $w_1$ from a Mallows model $\mathcal{M}_n(\phi_1, \pi_1)$, and with probability $w_2(= 1 - w_1)$ from a different model $\mathcal{M}_n(\phi_2, \pi_2)$.

**Informal Theorem.** *Given sufficiently many i.i.d samples drawn from a mixture of two Mallows models, we can learn the central permutations $\pi_1, \pi_2$ exactly and parameters $\phi_1, \phi_2, w_1, w_2$ up to $\epsilon$-accuracy in time* $\mathrm{poly}(n, (\min\{w_1, w_2\})^{-1}, \frac{1}{\phi_1(1-\phi_1)}, \frac{1}{\phi_2(1-\phi_2)}, \epsilon^{-1})$.

It is worth mentioning that, to the best of our knowledge, prior to this work even the question of identifiability was unresolved for a mixture of two Mallows models; given infinitely many i.i.d. samples generated from a mixture of two distinct Mallow models with parameters $\{w_1, \phi_1, \pi_1, w_2, \phi_2, \pi_2\}$ (with $\pi_1 \neq \pi_2$ or $\phi_1 \neq \phi_2$), could there be a different set of parameters $\{w_1', \phi_1', \pi_1', w_2', \phi_2', \pi_2'\}$ which explains the data just as well. Our result shows that this is not the case and the mixture is uniquely identifiable given polynomially many samples.

**Intuition and a Naïve First Attempt.** It is evident that having access to sufficiently many random samples allows one to learn a single Mallows model. Let the elements in the permutations be denoted as $\{e_1, e_2, \ldots, e_n\}$. In a single Mallows model, the probability of element $e_i$ going to position $j$ (for $j \in [n]$) drops off exponentially as one goes farther from the true position of $e_i$ [12]. So by assigning each $e_i$ the most frequent position in our sample, we can find the central ranking $\pi^*$.

The above mentioned intuition suggests the following clustering based approach to learn a mixture of two Mallows models — look at the distribution of the positions where element $e_i$ appears. If the distribution has 2 clearly separated "peaks" then they will correspond to the positions of $e_i$ in the central permutations. Now, dividing the samples according to $e_i$ being ranked in a high or a low position is likely to give us two pure (or almost pure) subsamples, each one coming from a single Mallows model. We can then learn the individual models separately. More generally, this strategy works when the two underlying permutations $\pi_1$ and $\pi_2$ are far apart which can be formulated as a *separation condition*.[2] Indeed, the above-mentioned intuition works only under strong separator conditions: otherwise, the observation regarding the distribution of positions of element $e_i$ is no longer true [3]. For example, if $\pi_1$ ranks $e_i$ in position $k$ and $\pi_2$ ranks $e_i$ in position $k + 2$, it is likely that the most frequent position of $e_i$ is $k+1$, which differs from $e_i$'s position in either permutations!

**Handling arbitrary permutations.** Learning mixture models under no separation requirements is a challenging task. To the best of our knowledge, the only polynomial time algorithm known is for the case of a mixture of a constant number of Gaussians [17, 18]. Other works, like the recent developments that use tensor based methods for learning mixture models without distance-based separation condition [19, 20, 21] still require non-degeneracy conditions and/or work for specific sub cases (e.g. spherical Gaussians).

These sophisticated tensor methods form a key component in our algorithm for learning a mixture of two Mallows models. This is non-trivial as learning over rankings poses challenges which are not present in other widely studied problems such as mixture of Gaussians. For the case of Gaussians, spectral techniques have been extremely successful [22, 16, 19, 21]. Such techniques rely on estimating the covariances and higher order moments in terms of the model parameters to detect structure and dependencies. On the other hand, in the mixture of Mallows models problem there is

*no "natural" notion of a second/third moment.* A key contribution of our work is defining analogous notions of moments which can be represented succinctly in terms of the model parameters. As we later show, this allows us to use tensor based techniques to get a good starting solution.

**Overview of Techniques.** One key difficulty in arguing about the Mallows model is the lack of closed form expressions for basic propositions like *"the probability that the $i$-th element of $\pi^*$ is ranked in position $j$."* Our first observation is that the distribution of a given element appearing at the top, i.e. the first position, behaves nicely. Given an element $e$ whose rank in the central ranking $\pi^*$ is $i$, the probability that a ranking sampled from a Mallows model ranks $e$ as the first element is $\propto \phi^{i-1}$. A length $n$ vector consisting of these probabilities is what we define as the *first moment vector* of the Mallows model. Clearly by sorting the coordinate of the first moment vector, one can recover the underlying central permutation and estimate $\phi$. Going a step further, consider any two elements which are in positions $i, j$ respectively in $\pi^*$. We show that the probability that a ranking sampled from a Mallows model ranks $\{i, j\}$ in (any of the 2! possible ordering of) the first two positions is $\propto f(\phi)\phi^{i+j-2}$. We call the $n \times n$ matrix of these probabilities as the *second moment matrix* of the model (analogous to the covariance matrix). Similarly, we define the 3rd moment tensor as the probability that any 3 elements appear in positions $\{1, 2, 3\}$. We show in the next section that in the case of a mixture of two Mallows models, the 3rd moment tensor defined this way has a rank-2 decomposition, with each rank-1 term corresponds to the first moment vector of each of two Mallows models. This motivates us to use tensor-based techniques to estimate the first moment vectors of the two Mallows models, thus learning the models' parameters.

The above mentioned strategy would work if one had access to infinitely many samples from the mixture model. But notice that the probabilities in the first-moment vectors decay exponentially, so by using polynomially many samples we can only recover a prefix of length $\sim \log_{1/\phi} n$ from both rankings. This forms the first part of our algorithm which outputs good estimates of the mixture weights, scaling parameters $\phi_1$, $\phi_2$ and prefixes of a certain size from both the rankings. Armed with $w_1$, $w_2$ and these two prefixes we next proceed to recover the full permutations $\pi_1$ and $\pi_2$. In order to do this, we take two new fresh batches of samples. On the first batch, we estimate the probability that element $e$ appears in position $j$ for all $e$ and $j$. On the second batch, which is noticeably larger than the first, we estimate the probability that $e$ appears in position $j$ *conditioned on a carefully chosen element $e^*$ appearing as the first element*. We show that this conditioning is *almost* equivalent to sampling from the same mixture model but with rescaled weights $w_1'$ and $w_2'$. The two estimations allow us to set a system of two linear equations in two variables: $f^{(1)}(e \to j)$ – the probability of element $e$ appearing in position $j$ in $\pi_1$, and $f^{(2)}(e \to j)$ — the same probability for $\pi_2$. Solving this linear system we find the position of $e$ in each permutation.

The above description contains most of the core ideas involved in the algorithm. We need two additional components. First, notice that the 3rd moment tensor is not well defined for triplets $(i, j, k)$, when $i, j, k$ are not all distinct and hence cannot be estimated from sampled data. To get around this barrier we consider a random partition of our element-set into 3 disjoint subsets. The actual tensor we work with consists only of triplets $(i, j, k)$ where the indices belong to different partitions. Secondly, we have to handle the case where tensor based-technique fails, i.e. when the 3rd moment tensor isn't full-rank. This is *a degenerate case*. Typically, tensor based approaches for other problems cannot handle such degenerate cases. However, in the case of the Mallows mixture model, we show that such a degenerate case provides a lot of useful information about the problem. In particular, it must hold that $\phi_1 \simeq \phi_2$, *and* $\pi_1$ and $\pi_2$ are fairly close — one is almost a cyclic shift of the other. To show this we use a characterization of the when the tensor decomposition is unique (for tensors of rank 2), and we handle such degenerate cases separately. Altogether, we find the mixture model's parameters with no non-degeneracy conditions.

**Lower bound under the pairwise access model.** Given that a single Mallows model can be learned using only pairwise comparisons, a very restricted access to each sample, it is natural to ask, *"Is it possible to learn a mixture of Mallows models from pairwise queries?"*. This next example shows that we cannot hope to do this even for a mixture of two Mallows models. Fix some $\phi$ and $\pi$ and assume our sample is taken using mixing weights of $w_1 = w_2 = \frac{1}{2}$ from the two Mallows models $\mathcal{M}_n(\phi, \pi)$ and $\mathcal{M}_n(\phi, \text{rev}(\pi))$, where $\text{rev}(\pi)$ indicates the reverse permutation (the first element of $\pi$ is the last of $\text{rev}(\pi)$, the second is the next-to-last, etc.) . Consider two elements, $e$ and $e'$. Using only pairwise comparisons, we have that it is just as likely to rank $e > e'$ as it is to rank $e' > e$ and so this case cannot be learned regardless of the sample size.

**3-wise queries.** We would also like to stress that our algorithm does not need full access to the sampled rankings and instead will work with access to certain 3-wise queries. Observe that the first part of our algorithm, where we recover the top elements in each of the two central permutations, only uses access to the top 3 elements in each sample. In that sense, we replace the pairwise query "do you prefer $e$ to $e'$?" with a 3-wise query: "what are your top 3 choices?" Furthermore, the second part of the algorithm (where we solve a set of 2 linear equations) can be altered to support 3-wise queries of the (admittedly, somewhat unnatural) form "if $e^*$ is your top choice, do you prefer $e$ to $e'$?" For ease of exposition, we will assume full-access to the sampled rankings.

**Future Directions.** Several interesting directions come out of this work. A natural next step is to generalize our results to learn a mixture of $k$ Mallows models for $k > 2$. We believe that most of these techniques can be extended to design algorithms that take $\text{poly}(n, 1/\epsilon)^k$ time. It would also be interesting to get algorithms for learning a mixture of $k$ Mallows models which run in time $\text{poly}(k, n)$, perhaps in an appropriate smoothed analysis setting [23] or under other non-degeneracy assumptions. Perhaps, more importantly, our result indicates that tensor based methods which have been very popular for learning problems, might also be a powerful tool for tackling ranking-related problems in the fields of machine learning, voting and social choice.

**Organization.** In Section 2 we give the formal definition of the Mallow model and of the problem statement, as well as some useful facts about the Mallow model. Our algorithm and its numerous subroutines are detailed in Section 3. In Section 4 we experimentally compare our algorithm with a popular EM based approach for the problem. The complete details of our algorithms and proofs are included in the supplementary material.

## 2 Notations and Properties of the Mallows Model

Let $U_n = \{e_1, e_2, \ldots, e_n\}$ be a set of $n$ distinct elements. We represent permutations over the elements in $U_n$ through their indices $[n]$. (E.g., $\pi = (n, n-1, \ldots, 1)$ represents the permutation $(e_n, e_{n-1}, \ldots, e_1)$.) Let $pos_\pi(e_i) = \pi^{-1}(i)$ refer to the position of $e_i$ in the permutation $\pi$. We omit the subscript $\pi$ when the permutation $\pi$ is clear from context. For any two permutations $\pi, \pi'$ we denote $d_{\mathsf{kt}}(\pi, \pi')$ as the Kendall-Tau distance [24] between them (number of pairwise inversions between $\pi, \pi'$). Given some $\phi \in (0, 1)$ we denote $Z_i(\phi) = \frac{1-\phi^i}{1-\phi}$, and partition function $Z_{[n]}(\phi) = \sum_\pi \phi^{d_{\mathsf{kt}}(\pi, \pi_0)} = \prod_{i=1}^n Z_i(\phi)$ (see Section 6 in the supplementary material).

**Definition 2.1.** *[Mallows model ($\mathcal{M}_n(\phi, \pi_0)$).] Given a permutation $\pi_0$ on $[n]$ and a parameter $\phi \in (0, 1)$,[4] a Mallows model is a permutation generation process that returns permutation $\pi$ w.p.*

$$\mathbf{Pr}(\pi) = \phi^{d_{\mathsf{kt}}(\pi, \pi_0)}/Z_{[n]}(\phi)$$

In Section 6 we show many useful properties of the Mallows model which we use repeatedly throughout this work. We believe that they provide an insight to Mallows model, and we advise the reader to go through them. We proceed with the main definition.

**Definition 2.2.** *[Mallows Mixture model $w_1\mathcal{M}_n(\phi_1, \pi_1) \oplus w_2\mathcal{M}_n(\phi_2, \pi_2)$.] Given parameters $w_1, w_2 \in (0, 1)$ s.t. $w_1 + w_2 = 1$, parameters $\phi_1, \phi_2 \in (0, 1)$ and two permutations $\pi_1, \pi_2$, we call a mixture of two Mallows models to be the process that with probability $w_1$ generates a permutation from $\mathcal{M}(\phi_1, \pi_1)$ and with probability $w_2$ generates a permutation from $\mathcal{M}(\phi_2, \pi_2)$.*

Our next definition is crucial for our application of tensor decomposition techniques.

**Definition 2.3.** *[Representative vectors.] The* representative vector *of a Mallows model is a vector where for every $i \in [n]$, the ith-coordinate is $\phi^{\text{pos}_\pi(e_i)-1}/Z_n$.*

The expression $\phi^{pos_\pi(e_i)-1}/Z_n$ is precisely the probability that a permutation generated by a model $\mathcal{M}_n(\phi, \pi)$ ranks element $e_i$ at the first position (proof deferred to the supplementary material). Given that our focus is on learning a mixture of two Mallows models $\mathcal{M}_n(\phi_1, \pi_1)$ and $\mathcal{M}_n(\phi_2, \pi_2)$, we denote $x$ as the representative vector of the first model, and $y$ as the representative vector of the latter. Note that retrieving the vectors $x$ and $y$ exactly implies that we can learn the permutations $\pi_1$ and $\pi_2$ and the values of $\phi_1, \phi_2$.

Finally, let $f\,(i \to j)$ be the probability that element $e_i$ goes to position $j$ according to mixture model. Similarly $f^{(1)}\,(i \to j)$ be the corresponding probabilities according to Mallows model $\mathcal{M}_1$ and $\mathcal{M}_2$ respectively. Hence, $f\,(i \to j) = w_1 f^{(1)}\,(i \to j) + w_2 f^{(2)}\,(i \to j)$.

**Tensors:** Given two vectors $u \in \mathbb{R}^{n_1}, v \in \mathbb{R}^{n_2}$, we define $u \otimes v \in R^{n_1 \times n_2}$ as the matrix $uv^T$. Given also $z \in \mathbb{R}^{n_3}$ then $u \otimes v \otimes z$ denotes the 3-tensor (of rank- 1) whose $(i, j, k)$-th coordinate is $u_i v_j z_k$. A tensor $T \in \mathbb{R}^{n_1 \times n_2 \times n_3}$ has a rank-$r$ decomposition if $T$ can be expressed as $\sum_{i \in [r]} u_i \otimes v_i \otimes z_i$ where $u_i \in \mathbb{R}^{n_1}, v_i \in \mathbb{R}^{n_2}, z_i \in \mathbb{R}^{n_3}$. Given two vectors $u, v \in \mathbb{R}^n$, we use $(u; v)$ to denote the $n \times 2$ matrix that is obtained with $u$ and $v$ as columns.

We now define first, second and third order statistics (frequencies) that serve as our proxies for the first, second and third order moments.

**Definition 2.4.** *[Moments] Given a Mallows mixture model, we denote for every $i, j, k \in [n]$*

- $P_i = \mathbf{Pr}\,(\mathrm{pos}\,(e_i) = 1)$ *is the probability that element $e_i$ is ranked at the first position*

- $P_{ij} = \mathbf{Pr}\,(\mathrm{pos}\,(\{e_i, e_j\}) = \{1, 2\})$*, is the probability that $e_i, e_j$ are ranked at the first two positions (in any order)*

- $P_{ijk} = \mathbf{Pr}\,(\mathrm{pos}\,(\{e_i, e_j, e_k\}) = \{1, 2, 3\})$ *is the probability that $e_i, e_j, e_k$ are ranked at the first three positions (in any order).*

For convenience, let $P$ represent the set of quantities $(P_i, P_{ij}, P_{ijk})_{1 \leq i < j < k \leq n}$. These can be estimated up to any inverse polynomial accuracy using only polynomial samples. The following simple, yet crucial lemma relates $P$ to the vectors $x$ and $y$, and demonstrates why these statistics and representative vectors are ideal for tensor decomposition.

**Lemma 2.5.** *Given a mixture $w_1 \mathcal{M}\,(\phi_1, \pi_1) \oplus w_2 \mathcal{M}\,(\phi_2, \pi_2)$ let $x, y$ and $P$ be as defined above.*

1. *For any $i$ it holds that $P_i = w_1 x_i + w_2 y_i$.*

2. *Denote $c_2(\phi) = \frac{Z_n(\phi)}{Z_{n-1}(\phi)} \frac{1+\phi}{\phi}$. Then for any $i \neq j$ it holds that $P_{ij} = w_1 c_2(\phi_1) x_i x_j + w_2 c_2(\phi_2) y_i y_j$.*

3. *Denote $c_3(\phi) = \frac{Z_n^2(\phi)}{Z_{n-1}(\phi) Z_{n-2}(\phi)} \frac{1 + 2\phi + 2\phi^2 + \phi^3}{\phi^3}$. Then for any distinct $i, j, k$ it holds that $P_{ijk} = w_1 c_3(\phi_1) x_i x_j x_k + w_2 c_3(\phi_2) y_i y_j y_k$.*

*Clearly, if $i = j$ then $P_{ij} = 0$, and if $i, j, k$ are not all distinct then $P_{ijk} = 0$.*

In addition, in Lemma 13.2 in the supplementary material we prove the bounds $c_2(\phi) = O(1/\phi)$ and $c_3(\phi) = O(\phi^{-3})$.

**Partitioning Indices:** Given a partition of $[n]$ into $S_a, S_b, S_c$, let $x^{(a)}, y^{(a)}$ be the representative vectors $x, y$ restricted to the indices (rows) in $S_a$ (similarly for $S_b, S_c$). Then the 3-tensor

$$T^{(abc)} \equiv (P_{ijk})_{i \in S_a, j \in S_b, k \in S_c} = w_1 c_3(\phi_1) x^{(a)} \otimes x^{(b)} \otimes x^{(c)} + w_2 c_3(\phi_2) y^{(a)} \otimes y^{(b)} \otimes y^{(c)}.$$

This tensor has a rank-2 decomposition, with one rank-1 term for each Mallows model. Finally for convenience we define the matrix $M = (x; y)$, and similarly define the matrices $M_a = (x^{(a)}; y^{(a)})$, $M_b = (x^{(b)}; y^{(b)})$, $M_c = (x^{(c)}; y^{(c)})$.

**Error Dependency and Error Polynomials.** Our algorithm gives an estimate of the parameters $w, \phi$ that we learn in the first stage, and we use these estimates to figure out the entire central rankings in the second stage. The following lemma essentially allows us to assume instead of estimations, we have access to the true values of $w$ and $\phi$.

**Lemma 2.6.** *For every $\delta > 0$ there exists a function $f(n, \phi, \delta)$ s.t. for every $n$, $\phi$ and $\hat{\phi}$ satisfying $|\phi - \hat{\phi}| < \frac{\delta}{f(n, \phi, \delta)}$ we have that the total-variation distance satisfies $\|\mathcal{M}\,(\phi, \pi) - \mathcal{M}\left(\hat{\phi}, \pi\right)\|_{\mathrm{TV}} \leq \delta$.*

For the ease of presentation, we do not optimize constants or polynomial factors in all parameters. In our analysis, we show how our algorithm is robust (in a polynomial sense) to errors in various statistics, to prove that we can learn with polynomial samples. However, the simplification when there are no errors (infinite samples) still carries many of the main ideas in the algorithm — this in fact shows the identifiability of the model, which was not known previously.

# 3 Algorithm Overview

---

**Algorithm 1** LEARN MIXTURES OF TWO MALLOWS MODELS, **Input:** a set $\mathcal{S}$ of $N$ samples from $w_1\mathcal{M}(\phi_1, \pi_1) \oplus w_2\mathcal{M}(\phi_2, \pi_2)$, Accuracy parameters $\epsilon, \epsilon_2$.

1. Let $\widehat{P}$ be the empirical estimate of $P$ on samples in $\mathcal{S}$.
2. Repeat $O(\log n)$ times:
   (a) Partition $[n]$ randomly into $S_a$, $S_b$ and $S_c$. Let $T^{(abc)} = \left(\widehat{P}_{ijk}\right)_{i \in S_a, j \in S_b, k \in S_c}$.
   (b) Run TENSOR-DECOMP from [25, 26, 23] to get a decomposition of $T^{(abc)} = u^{(a)} \otimes u^{(b)} \otimes u^{(c)} + v^{(a)} \otimes v^{(b)} \otimes v^{(c)}$.
   (c) If $\min\{\sigma_2(u^{(a)}; v^{(a)}), \sigma_2(u^{(b)}; v^{(b)}), \sigma_2(u^{(c)}; v^{(c)})\} > \epsilon_2$
      (In the *non-degenerate* case these matrices are far from being rank-1 matrices in the sense that their least singular value is bounded away from 0.)
      i. Obtain parameter estimates $(\widehat{w}_1, \widehat{w}_2, \widehat{\phi}_1, \widehat{\phi}_2$ and prefixes of the central rankings $\pi_1', \pi_2')$ from INFER-TOP-K$(\widehat{P}, M_a', M_b', M_c')$, with $M_i' = (u^{(i)}; v^{(i)})$ for $i \in \{a, b, c\}$.
      ii. Use RECOVER-REST to find the full central rankings $\widehat{\pi}_1, \widehat{\pi}_2$.
         Return SUCCESS and output $(\widehat{w}_1, \widehat{w}_2, \widehat{\phi}_1, \widehat{\phi}_2, \widehat{\pi}_1, \widehat{\pi}_2)$.
3. Run HANDLE DEGENERATE CASES $(\widehat{P})$.

---

Our algorithm (Algorithm 1) has two main components. First we invoke a decomposition algorithm [25, 26, 23] over the tensor $T^{(abc)}$, and retrieve approximations of the two Mallows models' representative vectors which in turn allow us to approximate the weight parameters $w_1, w_2$, scale parameters $\phi_1$, $\phi_2$, and the top few elements in each central ranking. We then use the inferred parameters to recover the entire rankings $\pi_1$ and $\pi_2$. Should the tensor-decomposition fail, we invoke a special procedure to handle such degenerate cases. Our algorithm has the following guarantee.

**Theorem 3.1.** *Let* $w_1\mathcal{M}(\phi_1, \pi_1) \oplus w_2\mathcal{M}(\phi_2, \pi_2)$ *be a mixture of two Mallows models and let* $w_{\min} = \min\{w_1, w_2\}$ *and* $\phi_{\max} = \max\{\phi_1, \phi_2\}$ *and similarly* $\phi_{\min} = \min\{\phi_1, \phi_2\}$. *Denote* $\epsilon_0 = \frac{w_{min}^2(1-\phi_{\max})^{10}}{16n^{22}\phi_{\max}^2}$. *Then, given any* $0 < \epsilon < \epsilon_0$, *suitably small* $\epsilon_2 = \mathrm{poly}(\frac{1}{n}, \epsilon, \phi_{min}, w_{min})$ *and* $N = \mathrm{poly}\left(n, \frac{1}{\min\{\epsilon, \epsilon_0\}}, \frac{1}{\phi_1(1-\phi_1)}, \frac{1}{\phi_2(1-\phi_2)}, \frac{1}{w_1}, \frac{1}{w_2}\right)$ *i.i.d samples from the mixture model, Algorithm 1 recovers, in poly-time and with probability* $\geq 1 - n^{-3}$, *the model's parameters with* $w_1, w_2, \phi_1, \phi_2$ *recovered up to* $\epsilon$-*accuracy.*

Next we detail the various subroutines of the algorithm, and give an overview of the analysis for each subroutine. The full analysis is given in the supplementary material.

**The TENSOR-DECOMP Procedure.** This procedure is a straight-forward invocation of the algorithm detailed in [25, 26, 23]. This algorithm uses spectral methods to retrieve the two vectors generating the rank-2 tensor $T^{(abc)}$. This technique works when all factor matrices $M_a = (x^{(a)}; y^{(a)})$, $M_b = (x^{(b)}; y^{(b)})$, $M_c = (x^{(c)}; y^{(c)})$ are well-conditioned. We note that any algorithm that decomposes non-symmetric tensors which have well-conditioned factor matrices, can be used as a black box.

**Lemma 3.2** (Full rank case). *In the conditions of Theorem 3.1, suppose our algorithm picks some partition* $S_a, S_b, S_c$ *such that the matrices* $M_a, M_b, M_c$ *are all well-conditioned — i.e. have* $\sigma_2(M_a), \sigma_2(M_b), \sigma_2(M_c) \geq \epsilon_2' \geq \mathrm{poly}(\frac{1}{n}, \epsilon, \epsilon_2, w_1, w_2)$ *then with high probability, Algorithm* TENSORDECOMP *of [25] finds* $M_a' = (u^{(a)}; v^{(a)}), M_b' = (u^{(b)}; v^{(b)}), M_c' = (u^{(c)}; v^{(c)})$ *such that for any* $\tau \in \{a, b, c\}$, *we have* $u^{(\tau)} = \alpha_\tau x^{(\tau)} + z_1^{(\tau)}$ *and* $v^{(\tau)} = \beta_\tau y^{(\tau)} + z_2^{(\tau)}$; *with* $\|z_1^{(\tau)}\|, \|z_2^{(\tau)}\| \leq \mathrm{poly}(\frac{1}{n}, \epsilon, \epsilon_2, w_{min})$ *and,* $\sigma_2(M_\tau') > \epsilon_2$ *for* $\tau \in \{a, b, c\}$.

**The INFER-TOP-K procedure.** This procedure uses the output of the tensor-decomposition to retrieve the weights, $\phi$'s and the representative vectors. In order to convert $u^{(a)}, u^{(b)}, u^{(c)}$ into an approximation of $x^{(a)}, x^{(b)}, x^{(c)}$ (and similarly with $v^{(a)}, v^{(b)}, v^{(c)}$ and $y^{(a)}, y^{(b)}, y^{(c)}$), we need to find a good approximation of the scalars $\alpha_a, \alpha_b, \alpha_c$. This is done by solving a certain linear system. This also allows us to estimate $\widehat{w}_1, \widehat{w}_2$. Given our approximation of $x$, it is easy to find $\phi_1$ and the top first elements of $\pi_1$ — we sort the coordinates of $x$, setting $\pi_1'$ to be the first elements in the sorted

vector, and $\phi_1$ as the ratio between any two adjacent entries in the sorted vector. We refer the reader to Section 8 in the supplementary material for full details. **The RECOVER-REST procedure.** The algorithm for recovering the remaining entries of the central permutations (Algorithm 2) is more involved.

---

**Algorithm 2** RECOVER-REST, **Input:** a set $\mathcal{S}$ of $N$ samples from $w_1 \mathcal{M}(\phi_1, \pi_1) \oplus w_2 \mathcal{M}(\phi_2, \pi_2)$, parameters $\hat{w}_1, \hat{w}_2, \hat{\phi}_1, \hat{\phi}_2$ and initial permutations $\hat{\pi}_1, \hat{\pi}_2$, and accuracy parameter $\epsilon$.

---

1. For elements in $\hat{\pi}_1$ and $\hat{\pi}_2$, compute representative vectors $\hat{x}$ and $\hat{y}$ using estimates $\hat{\phi}_1$ and $\hat{\phi}_2$.
2. Let $|\hat{\pi}_1| = r_1$, $|\hat{\pi}_2| = r_2$ and wlog $r_1 \geq r_2$.
   If there exists an element $e_i$ such that $pos_{\hat{\pi}_1}(e_i) > r_1$ and $pos_{\hat{\pi}_2}(e_i) < r_2/2$ (or in the symmetric case), then:
   Let $\mathcal{S}_1$ be the subsample with $e_i$ ranked in the first position.
   (a) Learn a single Mallows model on $\mathcal{S}_1$ to find $\hat{\pi}_1$. Given $\hat{\pi}_1$ use dynamic programming to find $\hat{\pi}_2$
3. Let $e_{i^*}$ be the first element in $\hat{\pi}_1$ having its probabilities of appearing in first place in $\pi_1$ and $\pi_2$ differ by at least $\epsilon$. Define $\hat{w}_1' = \left(1 + \frac{\hat{w}_2}{\hat{w}_1} \frac{\hat{y}(e_{i^*})}{\hat{x}(e_{i^*})}\right)^{-1}$ and $\hat{w}_2' = 1 - \hat{w}_1'$. Let $\mathcal{S}_1$ be the subsample with $e_{i^*}$ ranked at the first position.
4. For each $e_i$ that doesn't appear in either $\hat{\pi}_1$ or $\hat{\pi}_2$ and any possible position $j$ it might belong to
   (a) Use $\mathcal{S}$ to estimate $\hat{f}_{i,j} = \mathbf{Pr}(e_i$ goes to position $j)$, and $\mathcal{S}_1$ to estimate $\hat{f}(i \to j | e_{i^*} \to 1) = \mathbf{Pr}(e_i$ goes to position $j | e_{i^*} \mapsto 1)$.
   (b) Solve the system

$$\hat{f}(i \to j) = \hat{w}_1 f^{(1)}(i \to j) + \hat{w}_2 f^{(2)}(i \to j) \qquad (1)$$

$$\hat{f}(i \to j | e_{i^*} \to 1) = \hat{w}_1' f^{(1)}(i \to j) + \hat{w}_2' f^{(2)}(i \to j) \qquad (2)$$

5. To complete $\hat{\pi}_1$ assign each $e_i$ to position $\arg\max_j \{f^{(1)}(i \to j)\}$. Similarly complete $\hat{\pi}_2$ using $f^{(2)}(i \to j)$. Return the two permutations.

---

Algorithm 2 first attempts to find a pivot — an element $e_i$ which appears at a fairly high rank in one permutation, yet does not appear in the other prefix $\hat{\pi}_2$. Let $E_{e_i}$ be the event that a permutation ranks $e_i$ at the first position. As $e_i$ is a pivot, then $\mathbf{Pr}_{\mathcal{M}_1}(E_{e_i})$ is noticeable whereas $\mathbf{Pr}_{\mathcal{M}_2}(E_{e_i})$ is negligible. Hence, conditioning on $e_i$ appearing at the first position leaves us with a subsample in which all sampled rankings are generated from the first model. This subsample allows us to easily retrieve the rest of $\pi_1$. Given $\pi_1$, the rest of $\pi_2$ can be recovered using a dynamic programming procedure. Refer to the supplementary material for details.

The more interesting case is when no such pivot exists, i.e., when the two prefixes of $\pi_1$ and $\pi_2$ contain almost the same elements. Yet, since we invoke RECOVER-REST after successfully calling TENSOR-DECOMP , it must hold that the distance between the obtained representative vectors $\hat{x}$ and $\hat{y}$ is noticeably large. Hence some element $e_{i^*}$ satisfies $|\hat{x}(e_{i^*}) - \hat{y}(e_{i^*})| > \epsilon$, and we proceed by setting up a linear system. To find the complete rankings, we measure appropriate statistics to set up a system of linear equations to calculate $f^{(1)}(i \to j)$ and $f^{(2)}(i \to j)$ up to inverse polynomial accuracy. The largest of these values $\{f^{(1)}(i \to j)\}$ corresponds to the position of $e_i$ in the central ranking of $\mathcal{M}_1$.

To compute the values $\{f^{(r)}(i \to j)\}_{r=1,2}$ we consider $f^{(1)}(i \to j | e_{i^*} \to 1)$ – the probability that $e_i$ is ranked at the $j$th position conditioned on the element $e_{i^*}$ ranking first according to $\mathcal{M}_1$ (and resp. for $\mathcal{M}_2$). Using $w_1'$ and $w_2'$ as in Algorithm 2, it holds that

$$\mathbf{Pr}(e_i \to j | e_{i^*} \to 1) = w_1' f^{(1)}(i \to j | e_{i^*} \to 1) + w_2' f^{(2)}(i \to j | e_{i^*} \to 1).$$

We need to relate $f^{(r)}(i \to j | e_{i^*} \to 1)$ to $f^{(r)}(i \to j)$. Indeed Lemma 10.1 shows that $\mathbf{Pr}(e_i \to j | e_{i^*} \to 1)$ is an *almost* linear equations in the two unknowns. We show that if $e_{i^*}$ is ranked above $e_i$ in the central permutation, then for some small $\delta$ it holds that

$$\mathbf{Pr}(e_i \to j | e_{i^*} \to 1) = w_1' f^{(1)}(i \to j) + w_2' f^{(2)}(i \to j) \pm \delta$$

We refer the reader to Section 10 in the supplementary material for full details.

**The HANDLE-DEGENERATE-CASES procedure.** We call a mixture model $w_1\mathcal{M}(\phi_1,\pi_1) \oplus w_2\mathcal{M}(\phi_2,\pi_2)$ *degenerate* if the parameters of the two Mallows models are equal, and the edit distance between the prefixes of the two central rankings is at most two i.e., by changing the positions of at most two elements in $\pi_1$ we retrieve $\pi_2$. We show that unless $w_1\mathcal{M}(\phi_1,\pi_1) \oplus w_2\mathcal{M}(\phi_2,\pi_2)$ is degenerate, a random partition $(S_a, S_b, S_c)$ is likely to satisfy the requirements of Lemma 3.2 (and TENSOR-DECOMP will be successful). Hence, if TENSOR-DECOMP repeatedly fail, we deduce our model is indeed degenerate. To show this, we characterize the uniqueness of decompositions of rank 2, along with some very useful properties of random partitions. In such degenerate cases, we find the two prefixes and then remove the elements in the prefixes from $U$, and recurse on the remaining elements. We refer the reader to Section 9 in the supplementary material for full details.

## 4 Experiments

**Goal.** The main contribution of our paper is devising an algorithm that *provably* learns any mixture of two Mallows models. But could it be the case that the previously existing heuristics, even though they are unproven, still perform well in practice? We compare our algorithm to existing techniques, to see if, and under what settings our algorithm outperforms them.

**Baseline.** We compare our algorithm to the popular EM based algorithm of [5], seeing as EM based heuristics are the most popular way to learn a mixture of Mallows models. The EM algorithm starts with a random guess for the two central permutations. At iteration $t$, EM maintains a guess as to the two Mallows models that generated the sample. First (expectation step) the algorithm assigns a weight to each ranking in our sample, where the weight of a ranking reflects the probability that it was generated from the first or the second of the current Mallows models. Then (the maximization step) the algorithm updates its guess of the models' parameters based on a local search – minimizing the average distance to the weighted rankings in our sample. We comment that we implemented only the version of our algorithm that handles non-degenerate cases (more interesting case). In our experiment the two Mallows models had parameters $\phi_1 \neq \phi_2$, so our setting was never degenerate.

**Setting.** We ran both the algorithms on synthetic data comprising of rankings of size $n = 10$. The weights were sampled u.a.r from $[0, 1]$, and the $\phi$-parameters were sampled by sampling $\ln(1/\phi)$ u.a.r from $[0, 5]$. For $d$ ranging from 0 to $\binom{n}{2}$ we generated the two central rankings $\pi_1$ and $\pi_2$ to be within distance $d$ in the following manner. $\pi_1$ was always fixed as $(1, 2, 3, \ldots, 10)$. To describe $\pi_2$, observe that it suffices to note the number of inversion between 1 and elements $2, 3, ..., 10$; the number of inversions between 2 and $3, 4, ..., 10$ and so on. So we picked u.a.r a non-negative integral solution to $x_1 + \ldots + x_n = d$ which yields a feasible permutation and let $\pi_2$ be the permutation that it details. Using these models' parameters, we generated $N = 5 \cdot 10^6$ random samples.

**Evaluation Metric and Results.** For each value of $d$, we ran both algorithms 20 times and counted the fraction of times on which they returned the true rankings that generated the sample. The results of the experiment for rankings of size $n = 10$ are in Table 1. Clearly, the closer the two centrals rankings are to one another, the worst EM performs. On the other hand, our algorithm is able to recover the true rankings even at very close distances. As the rankings get slightly farther, our algorithm recovers the true rankings all the time. We comment that similar performance was observed for other values of $n$ as well. We also comment that our algorithm's runtime was reasonable (less than 10 minutes on a 8-cores Intel x86_ 64 computer). Surprisingly, our implementation of the EM algorithm typically took much longer to run — due to the fact that it simply did not converge.

| distance between rankings | success rate of EM | success rate of our algorithm |
|---|---|---|
| 0 | 0% | 10% |
| 2 | 0% | 10% |
| 4 | 0% | 40% |
| 8 | 10% | 70% |
| 16 | 30% | 60 % |
| 24 | 30% | 100% |
| 30 | 60% | 100% |
| 35 | 60% | 100% |
| 40 | 80% | 100% |
| 45 | 60% | 100% |

Table 1: Results of our experiment.

## Footnotes

*This work was supported in part by NSF grants CCF-1101215, CCF-1116892, the Simons Institute, and a Simons Foundation Postdoctoral fellowhsip. Part of this work was performed while the 3rd author was at the Simons Institute for the Theory of Computing at the University of California, Berkeley and the 4th author was at CMU.

[1]In fact, it was shown [1] that this model is the result of the following simple (inefficient) algorithm: rank every pair of elements randomly and independently s.t. with probability $\frac{1}{1+\phi}$ they agree with $\pi^*$ and with probability $\frac{\phi}{1+\phi}$ they don't; if all $\binom{n}{2}$ pairs agree on a single ranking – output this ranking, otherwise resample.

[2]Identifying a permutation $\pi$ over $n$ elements with a $n$-dimensional vector $(\pi(i))_i$, this separation condition can be roughly stated as $\|\pi_1 - \pi_2\|_\infty = \tilde{\Omega}\left((\min\{w_1, w_2\})^{-1} \cdot (\min\{\log(1/\phi_1), \log(1/\phi_2)\})^{-1}\right)$.

[3]Much like how other mixture models are solvable under separation conditions, see [14, 15, 16].

[4]It is also common to parameterize using $\beta \in \mathbb{R}^+$ where $\phi = e^{-\beta}$. For small $\beta$ we have $(1 - \phi) \approx \beta$.

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
