[Supplementary Material]

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

# 5  Acknowledgements

We would like to thank Ariel Procaccia for bringing to our attention various references to Mallows model in social choice theory.

# 6  Properties of the Mallows Model

In this section, we outline some of the properties of the Mallows model. Some of these properties were already shown before (see [27]), but we add them in this appendix for completion. Our algorithm and its analysis rely heavily on these properties.

**Notation.** Given a Mallows model $\mathcal{M}_n(\phi, \pi_0)$ we denote $Z_n = \frac{1-\phi^n}{1-\phi}$, and we denote $Z_{[n]}$ as the sum all weights of all permutations: $Z_{[n]} = \sum_\pi \phi^{d_{\mathsf{kt}}(\pi, \pi_0)}$. Given an element $e$, we abuse notation and denote by $\pi \setminus e$ the permutation we get by omitting the element $e$ (projecting $\pi$ over all elements but $e$). The notation $\pi = (e, \sigma)$ denotes a permutation whose first element is $e$ and elements 2 through $n$ are as given by the permutation over $n-1$ elements $\sigma$.

The first property shows that for any element $e$, conditioning on $e$ being ranked at the first position results in a reduced Mallows model.

**Lemma 6.1.** *Let $\mathcal{M}(\phi, \pi)$ be a Mallows model over $[n]$. For any $i$, the conditional distribution (given that $i$ is ranked at position 1) of rankings over $[n] \setminus \{i\}$, i.e. $\mathbf{Pr}(\pi | \pi(i) = 1)$ is the same as that of $\mathcal{M}(\phi, \pi \setminus i)$.*

The above lemma can be extended to conditioning on prefixes as follows.

**Lemma 6.2.** *Let $\mathcal{M}(\phi, \pi)$ be a Mallows model over $[n]$. For any prefix $I$ of $\pi$, the marginal distribution of rankings over $[n] \setminus I$ is the same as that of $\mathcal{M}(\phi, \pi \setminus I)$.*

The following lemma describe a useful trick that allows us to simulate the addition of another element that is added to the start of the central ranking $\pi$, using the knowledge of $\phi$. This will be particularly useful to simplify certain degenerate cases.

**Lemma 6.3.** *Let $\mathcal{M}(\phi, \pi)$ be a Mallows model over $[n]$. Given oracle access to $\mathcal{M}(\phi, \pi)$ and a new element $e_0 \notin [n]$ we can efficiently simulate an oracle access to $\mathcal{M}(\phi, (e_0, \pi))$.*

## 6.1  Proofs of Lemmas 6.1, 6.2, 6.3

**Observation.** All of the properties we state and prove in this appendix are based on the following important observation. Given two permutations $\pi$ and $\pi'$, denote the first element in $\pi$ as $e_1$. Then we have that

$$\#\text{pairs } (e_1, e_i)_{i \neq 1} \text{ that } \pi, \pi' \text{ disagree on} = (\text{position of } e_1 \text{ in } \pi') - 1 = pos_{\pi'}(e_1) - 1$$

The same holds for the last element, denoted $e_n$, only using the distance between $pos_{\pi'}(e_n)$ and the $n$th-position (i.e., $n - pos_{\pi'}(e_n)$).

We begin by characterizing $Z_{[n]}$.

**Property 6.4.** *For every $n$ and any $\pi_0 \in S_n$ we have that $Z_{[n]} = \sum_\pi \phi^{d_{\mathsf{kt}}(\pi, \pi_0)} = \prod_{i=1}^n Z_i = \prod_i \left( \sum_{j=0}^{i=1} \phi^j \right)$.*

*Proof.* By induction on $n$. For $n = 1$ there's a single permutation over the set $\{1\}$ and $Z_1 = 1$. For any $n > 1$, given a permutation over $n$ elements $\pi \in S_n$, denote its first element as $e_\pi$. Based on our observation, we have that

$$d_{\mathsf{kt}}(\pi, \pi_0) = \#\text{swaps involving } e_\pi + d_{\mathsf{kt}}(\pi \setminus e_\pi, \pi_0 \setminus e_\pi) = (pos_{\pi_0}(e_\pi) - 1) + d_{\mathsf{kt}}(\pi \setminus e_\pi, \pi_0 \setminus e_\pi)$$

And so we have

$$Z_{[n]} = \sum_\pi \phi^{d_{\mathsf{kt}}(\pi, \pi_0)} = \sum_{j=1}^n \sum_{\{\pi : e_\pi \text{ is the } j\text{th elements in } \pi_0\}} \phi^{d_{\mathsf{kt}}(\pi, \pi_0)}$$

$$= \sum_{j=1}^{n} \sum_{\{\pi : e_\pi \text{ is the } j\text{th elements in } \pi_0\}} \phi^{j-1} \phi^{d_{\mathsf{kt}}(\pi \backslash e_\pi, \pi_0 \backslash e_\pi)}$$

$$= \sum_{j=0}^{n-1} \phi^j \sum_{\pi \in S_{n-1}} \phi^{d_{\mathsf{kt}}(\pi, \pi_0^{-j})}$$

$$\stackrel{\text{induction}}{=} \sum_{j=0}^{n-1} \phi^j \left( \prod_{i=1}^{n-1} Z_i \right) = \left( \prod_{i=1}^{n-1} Z_i \right) Z_n = \prod_{i=1}^{n} Z_i$$

where $\pi_0^{-j}$ denotes the permutation we get by omitting the $j$th element from $\pi_0$. $\qquad \square$

Observe that the proof essentially shows how to generate a random ranking from a Mallows model. What we in fact showed is that the given a permutation $\pi = (e, \pi \backslash e)$ we have that

$$\mathbf{Pr}[\pi] = \tfrac{1}{Z_{[n]}} \phi^{(pos_{\pi_0}(e)-1)+d_{\mathsf{kt}}(\pi \backslash e, \pi_0 \backslash e)} = \frac{\phi^{(pos_{\pi_0}(e)-1)}}{Z_n} \cdot \frac{\phi^{d_{\mathsf{kt}}(\pi \backslash e, \pi_0 \backslash e)}}{Z_{1:(n-1)}}$$

And so, to generate a random permutation using $\pi_0$: place the $j$th elements of $\pi_0$ at the first position w.p. $\propto \phi^{j-1}$, and recourse over the truncated permutation. $\pi_0 \backslash e_1$ to find the rest of the permutation (positions $1, 2, \ldots, j-1, j+1, \ldots, n$). This proves Lemma 6.1.

Note the symmetry between $\pi$ and $\pi_0$ in defining the weight of $\pi$. Therefore, denoting $e_1$ as the element $\pi_0$ ranks at the first position, we have that

$$d_{\mathsf{kt}}(\pi, \pi_0) = \#\text{swaps involving } e_1 + d_{\mathsf{kt}}(\pi \backslash e_1, \pi_0 \backslash e_1) = (i-1) + d_{\mathsf{kt}}(\pi \backslash e_1, \pi_0 \backslash e_1)$$

and so, the probability of permutation $\pi$ in which $e_1$ is ranked at position $j$ and the rest of the permutation is as a given permutation $\sigma$ over $n-1$ elements is:

$$\mathbf{Pr}[\pi] = \tfrac{1}{Z_{[n]}} \phi^{(j-1)+d_{\mathsf{kt}}(\pi \backslash e_1, \pi_0 \backslash e_1)} = \frac{\phi^{(j-1)}}{Z_n} \cdot \frac{\phi^{d_{\mathsf{kt}}(\pi \backslash e_1, \pi_0 \backslash e_1)}}{Z_{1:(n-1)}}$$

So, an alternative way to generate a random permutation using $\pi_0$ is to rank element $e_1$ at position $j$ w.p. $\propto \phi^{j-1}$ and then to recourse over the truncated permutation $\pi_0 \backslash e_1$. Repeating this argument for each element in a given prefix $I$ of $\pi_0$ proves Lemma 6.2.

Observe that the algorithms the generate a permutation for a given Mallows model also allow us to simulate a random sample from a Mallows model over $n+1$ elements. That is, given $\pi_0$, we can introduce a new element $e_0$ and denote $\pi_0' = (e_0, \pi_0)$. Now, to sample from a Mallows model centered at $\pi_0'$ all we need is to pick the position of $e_0$ (moving it to position $j$ w.p. $\phi^{j-1}/Z_{n+1}$), then sampling from original Mallows model. This proves Lemma 6.3.

## 6.2  Total Variation Distance

In this subsection, our goal is to prove Lemma 2.6. Namely, we aim to show that given $\phi$, for every $\delta > 0$ we can pick any $\hat{\phi}$ sufficiently close to $\phi$, and have that the total variation distance between the two models $\mathcal{M}(\phi, \pi_0)$ and $\mathcal{M}\left(\hat{\phi}, \pi_0\right)$ is at most $\delta$.

*Proof of Lemma 2.6.* First, denote $\phi = e^{-\beta}$ and $\hat{\phi} = e^{-\hat{\beta}}$. And so it holds that

$$|\beta - \hat{\beta}| = |\ln(1/\phi) - \ln(1/\hat{\phi})| = |\ln(\hat{\phi}/\phi)| \leq |\ln(1 + \tfrac{|\phi - \hat{\phi}|}{\phi_{\min}})| \leq \tfrac{|\phi - \hat{\phi}|}{\phi_{\min}}$$

assuming some global lower bound $\phi_{\min}$ on $\phi, \hat{\phi}$.

Observe that for every $\pi$ we have that

$$\phi^{d_{\mathsf{kt}}(\pi, \pi_0)} = \exp(-\beta d_{\mathsf{kt}}(\pi, \pi_0)) = \exp(-\hat{\beta} d_{\mathsf{kt}}(\pi, \pi_0)) \exp(-(\beta - \hat{\beta}) d_{\mathsf{kt}}(\pi, \pi_0)) \leq e^{\frac{1}{2} n^2 |\beta - \hat{\beta}|} \hat{\phi}^{d_{\mathsf{kt}}(\pi, \pi_0)}$$

**Algorithm 3** LEARN MIXTURES OF TWO MALLOWS MODELS, **Input:** a set $\mathcal{S}$ of $N$ samples from $w_1 \mathcal{M}(\phi_1, \pi_1) \oplus w_2 \mathcal{M}(\phi_2, \pi_2)$, Accuracy parameters $\epsilon, \epsilon_2$.

1. Set threshold $\epsilon_2 = f_2(\epsilon)$.
2. Let $\widehat{P}$ be the empirical estimate of $P$ on samples in $\mathcal{S}$.
3. Run $O(\log n)$ times
   (a) Partition $[n]$ randomly into $S_a$, $S_b$ and $S_c$.
   (b) Set $T^{(abc)} = \left( \widehat{P}_{ijk} \right)_{i \in S_a, j \in S_b, k \in S_c}$.
   (c) Run TENSOR-DECOMP as in Theorem 4.2 of([25]) to get a decomposition of $T_{abc} = u^{(a)} \otimes u^{(b)} \otimes u^{(c)} + v^{(a)} \otimes v^{(b)} \otimes v^{(c)}$.
   (d) Let $M_a' = (u^{(a)}; v^{(a)})$, $M_b' = (u^{(b)}; v^{(b)})$, $M_c' = (u^{(c)}; v^{(c)})$.
   (e) If $\min(\sigma_2(M_a'), \sigma_2(M_b'), \sigma_2(M_c')) \geq \epsilon_2$,
       i. $(\widehat{w}_1, \widehat{w}_2, \widehat{\phi}_1, \widehat{\phi}_2, \pi_1', \pi_2') \leftarrow$ INFER-TOP-K$(\widehat{P}, M_a', M_b', M_c')$.
       ii. $(\widehat{\pi}_1, \widehat{\pi}_2) \leftarrow$ RECOVER-REST$(\mathcal{S}, \widehat{w}_1, \widehat{w}_2, \widehat{\phi}_1, \widehat{\phi}_2, \pi_1', \pi_2', \epsilon_2/\sqrt{2n})$.
          Return SUCCESS and output $(\widehat{w}_1, \widehat{w}_2, \widehat{\phi}_1, \widehat{\phi}_2, \widehat{\pi}_1, \widehat{\pi}_2)$.
   (f) Else if $\sigma_2(M_a') < \epsilon_2$ and $\sigma_2(M_b') \geq \epsilon_2$, and $\sigma_2(M_c') \geq \epsilon_2$ (or other symmetric cases), let $p^{(a)} = \left( \widehat{P}_i \right)_{i \in S_a}$.
       $\widehat{\phi} \leftarrow$ ESTIMATE-PHI$(p^{(a)})$.
   (g) Else $\widehat{\phi} = $ median $\left(\text{ESTIMATE-PHI}(p^{(a)}), \text{ESTIMATE-PHI}(p^{(b)}), \text{ESTIMATE-PHI}(p^{(c)})\right)$.
   (h) Else, (at least two of the three matrices $M_a'$, $M_b'$, $M_c'$ are essentially rank-1) let $\tau \in \{a, b, c\}$ denote a matrix $M_\tau'$ s.t. $\sigma_2(M_\tau') < \epsilon_2$, and let $p^{(\tau)} = (\widehat{P}_i)_{i \in S_\tau}$.
       $\widehat{\phi} \leftarrow$ ESTIMATE-PHI$(p^{(\tau)})$.
4. Run HANDLE-DEGENERATE-CASE$(\widehat{P}, \hat{\phi}, \epsilon)$.

Summing over all permutation (and replacing the role of $\phi$ and $\hat{\phi}$) we have also that $\sum_\pi \phi^{d_{kt}(\pi, \pi_0)} \geq e^{-\frac{1}{2} n^2 |\beta - \hat{\beta}|} \sum_\pi \hat{\phi}^{d_{kt}(\pi, \pi_0)}$. Let $p_\pi$ (resp. $\hat{p}_\pi$) denote the probability of sampling the permutation $\pi$ from a Mallows model $\mathcal{M}(\phi, \pi_0)$ (resp. $\mathcal{M}\left(\hat{\phi}, \pi_0\right)$). It follows that for every $\pi$ we have

$$p_\pi = \frac{\phi^{d_{kt}(\pi, \pi_0)}}{\sum_{\pi'} \phi^{d_{kt}(\pi', \pi_0)}} \leq e^{n^2 |\beta - \hat{\beta}|} \frac{\hat{\phi}^{d_{kt}(\pi, \pi_0)}}{\sum_{\pi'} \hat{\phi}^{d_{kt}(\pi', \pi_0)}} = e^{n^2 |\beta - \hat{\beta}|} \hat{p}_\pi$$

and similarly, $\hat{p}_\pi \leq e^{n^2 |\beta - \hat{\beta}|} p_\pi$.

Therefore, assuming that $|\beta - \hat{\beta}|$ is sufficiently small, and using the fact that $|1 - e^x| \leq 2|x|$ for $x \in (-\frac{1}{2}, \frac{1}{2})$, then we have

$$\begin{aligned}
\left\| \mathcal{M}(\phi, \pi) - \mathcal{M}\left(\hat{\phi}, \pi\right) \right\|_{\text{TV}} &= \frac{1}{2} \sum_\pi |p_\pi - \hat{p}_\pi| \\
&= \frac{1}{2} \sum_\pi p_\pi \left| 1 - \frac{\hat{p}_\pi}{p_\pi} \right| \leq \frac{1}{2} \sum_\pi 2 p_\pi n^2 |\beta - \hat{\beta}| = \frac{n^2}{\phi_{\min}} |\phi - \hat{\phi}|
\end{aligned}$$

It follows that in order to bound the total variation distance by $\delta$ we need to have $\phi$ and $\hat{\phi}$ close up to a factor of $\delta \cdot \phi_{\min}/n^2$. □

# 7 Algorithm and Subroutines

We now describe the algorithm and its subroutines in full detail. These will be followed by the analysis of the algorithms and proof of correctness in the following sections. Broadly speaking, our algorithm (Algorithm 1) has two main components.

**Retrieving the Top Elements and Parameters.** In the first part we use spectral methods to recover elements which have a good chance of appearing in the first position. The algorithm tries

**Algorithm 4** INFER-TOP-K, **Input:** $\widehat{P}, M_a' = (u^{(a)}; v^{(a)}), M_b' = (u^{(b)}; v^{(b)}), M_c' = (u^{(c)}; v^{(c)})$.

1. Let $\hat{P}_a = \hat{P}(i \in a)$

2. Set $(\alpha_a, \beta_a)^T = (M_a')^\dagger \hat{P}_a$
   $(\alpha_b, \beta_b)^T = (M_b')^\dagger \hat{P}_b$
   $(\alpha_c, \beta_c)^T = (M_c')^\dagger \hat{P}_c$.

3. Set $\hat{w_1} = \|\alpha_a u^{(a)}\|_1 + \|\alpha_b u^{(b)}\|_1 + \|\alpha_c u^{(c)}\|_1$, $\hat{w_2} = 1 - \hat{w_1}$.

4. Let $u = \left( \frac{\alpha_a}{w_1} u^{(a)}, \frac{\alpha_b}{w_1} u^{(b)}, \frac{\alpha_c}{w_1} u^{(c)} \right)$.
   $v = \left( \frac{\beta_a}{w_2} v^{(a)}, \frac{\beta_b}{w_2} v^{(b)}, \frac{\beta_c}{w_2} v^{(c)} \right)$.

5. Sort the vectors $u$ and $v$ in decreasing order, i.e., $U \leftarrow \text{SORT}(u)$, $V \leftarrow \text{SORT}(v)$.

6. $\hat{\phi}_1 = \frac{U_2}{U_1}$ and $\hat{\phi}_2 = \frac{V_2}{V_1}$.

7. Define $\gamma = \frac{(1 - \hat{\phi}_{\max})^2}{4n\hat{\phi}_{\max}}$. Let $r_1 = \log_{1/\hat{\phi}_1} \left( \frac{n^{10}}{w_{\min}^2 \gamma^2} \right)$ and $r_2 = \log_{1/\hat{\phi}_2} \left( \frac{n^{10}}{w_{\min}^2 \gamma^2} \right)$.

8. Output $\pi_1'$ to be the first $r_1$ ordered elements according to $U$ and $\pi_2'$ to be the first $r_2$ ordered elements according to $V$.

---

$O(\log n)$ different random partitions $S_a, S_b, S_c$, and constructs the tensor $T^{(abc)}$ from the samples as described in step 3(b). We then try to find a rank-2 decomposition of the tensor using a black-box algorithm for decomposing non-symmetric tensors. While we use the algorithm of [25] here, we can use the more practically efficient algorithm of Jennrich [23], or other power-iteration methods that are suitably modified to handle non-symmetric tensors.

These algorithms work when the factor matrices $M_a, M_b, M_c$ have polynomially bounded condition number (in other words their second largest singular values $\sigma_2(\cdot)$ is lower bounded by a polynomial in the input parameters) — in such cases the tensor $T^{(abc)}$ has a unique rank-2 decomposition. If this condition holds for any of the random partitions, then one can recover the top few elements of both $\pi_1$ and $\pi_2$ correctly. In addition, we can also infer the parameters $w$'s and $\phi$'s to good accuracy $\epsilon$ (corresponding to INFER-TOP-K (Algorithm 4). This is detailed in section 8.

If any random partition $S_a, S_b, S_c$ fails to produce a tensor $T^{(abc)}$ with well-conditioned factor matrices, then we are already in a special case. We show that in this case, the scaling parameters $\phi_1 \approx \phi_2$ with high probability. We exploit the random choice of the partition to make this argument (see Lemma 9.1). However, we still need to find the top few elements of the permutations and the weights. If all these $O(\log n)$ random partitions fail, then we show that we are in the *Degenerate case* that we handle separately; we describe a little later. Otherwise, if at least one of the random partitions succeeds, then we have estimated the scaling parameters, the mixing weights and the top few elements of both permutations.

**Recovering Rest of the Elements.** The second part of the algorithm (corresponding to RECOVER-REST) takes the inferred parameters and the initial prefixes as input and uses this information to recover the entire rankings $\pi_1$ and $\pi_2$. This is done by observing that the probability of an element $e_i$ going to position $j$ can be written as a weighted combination of the corresponding probabilities under $\pi_1$ and $\pi_2$. In addition, as mentioned in Section 2, the reduced distribution obtained by conditioning on a particular element $e_j$ going to position 1 is again a mixture of two Mallows models with the same parameters. Hence, by conditioning on a particular element which appears in the initial learned prefix, we get a system of linear equations which can be used to infer the probability of every other element $e_i$ going to position $j$ in both $\pi_1$ and $\pi_2$. This will allow us to infer the entire rankings.

**Degenerate Cases.** In the case when none of the random partition produces a tensor which has well-conditioned factor matrices (or alternately, a unique rank-2 decomposition), the instance is a very special instance, that we term *degenerate*. The additional subroutine (HANDLE-DEGENERATE-CASE) takes care of such degenerate instances. Before we do so, we introduce some notation to describe these degenerate cases.

*Notation.* Define $L_\epsilon = \{e_i : P_i \geq \epsilon\}$. If $\epsilon$ not stated explicitly $L$ refers to $L_{\sqrt{\epsilon}}$ where $\epsilon$ is the accuracy required in Theorem 3.1.

Now we have the following definition that helps us formally define the degenerate case.

**Definition 7.1** (Bucketing by relative positions). *For every $\ell \in \mathbb{Z}$, let $B_\ell = \{e_i \in L : pos_{\pi_1}(e_i) - pos_{\pi_2}(e_i) = \ell\}$. Further let $\ell^*$ be the majority bucket for the elements in $L$.*

We call a mixture model $w_1 \mathcal{M}(\phi_1, \pi_1) \oplus w_2 \mathcal{M}(\phi_2, \pi_2)$ as degenerate if except for at most 2 elements, all the elements in $L$ fall into the majority bucket. In other words, $|\ell^*| \geq |L| - 2$. Intuitively, in this case one of the partitions $S_a, S_b, S_c$ constructed by the algorithm will have their corresponding $u$ and $v$ vectors as parallel to each other and hence the tensor method will fail. We show that when this happens, it can be detected and in fact this case provides useful information about the model parameters. More specifically, we show that in a degenerate case, $\phi_1$ will be almost equal to $\phi_2$ and the two rankings will be aligned in a couple of very special configurations (see Section 9). Procedure HANDLE-DEGENERATE-CASE is designed to recover the rankings in such scenarios.

# 8 Retrieving the Top elements

Here we show how the first stage of the algorithm i.e. *steps (a)-(e.i)* manages to recover the top few elements of both rankings $\pi_1$ and $\pi_2$ and also estimate the parameters $\phi_1, \phi_2, w_1, w_2$ up to accuracy $\epsilon$. We first show that if $M_a, M_b, M_c$ have non-negligible minimum singular values (at least $\epsilon_2'$ as in Lemma 8.1), then the decomposition is unique, and hence we can recover the top few elements and parameters from INFER TOP-K. Otherwise, we show that if this procedure did not work for all $O(\log n)$ iterations, we are in *the degenerate case* (Lemma 9.1 and Lemma 9.6), and handle this separately.

For the sake of analysis, we denote by $\gamma_{\min}$ the smallest length of the vectors in the partition i.e. $\gamma_{\min} = \min_{\tau \in \{a,b,c\}} \min \{\|x^{(\tau)}\|, \|y^{(\tau)}\|\}$. Lemma 9.10 shows that with high probability $\gamma_{\min} \geq \phi_{\min}^{C \log n}(1 - \phi)$ for some large constant $C$.

The following lemma shows that when $M_a, M_b, M_c$ are well-conditioned, Algorithm TENSORDE-COMP finds a decomposition close to the true decomposition up to scaling. This Lemma essentially follows from the guarantees of the Tensor Decomposition algorithm in [25]. It also lets us conclude that $\sigma_2(M_a'), \sigma_2(M_b'), \sigma_2(M_c')$ are all also large enough. Hence, these singular values of the matrices $M_a', M_b', M_c'$ that we obtain from TENSOR-DECOMP algorithm can be tested to check if this step worked.

**Lemma 8.1** (Decomposition guarantees). *In the conditions of Theorem 3.1, suppose there exists a partition $S_a, S_b, S_c$ such that the matrices $M_a = (x^{(a)}; y^{(a)}), M_b = (x^{(b)}; y^{(b)})$ and $M_c = (x^{(c)}; y^{(c)})$ are well-conditioned i.e. $\sigma_2(M_a), \sigma_2(M_b), \sigma_2(M_c) \geq \epsilon_2'$, then with high probability, Algorithm TENSORDECOMP finds $M_a' = (u^{(a)}; v^{(a)}), M_b' = (u^{(b)}; v^{(b)}), M_c' = (u^{(c)}; v^{(c)})$ such that*

1. *For $\tau \in \{a, b, c\}$, we have $u^{(\tau)} = \alpha_a x^{(\tau)} + z_1^{(\tau)}$ and $v^{(\tau)} = \beta_a y^{(\tau)} + z_2^{(\tau)}$ where $\|z_1^{(\tau)}\|, \|z_2^{(\tau)}\| \leq \vartheta_{8.1}(n, \epsilon, \epsilon_2, w_{min})$*

2. *$\sigma_2(M_a') \geq \gamma_{min}(\epsilon_2' - \vartheta_{8.1})$ (similarly for $M_b', M_c'$).*

*where $\vartheta_{8.1}$ is a polynomial function $\vartheta_{8.1} = \min \left\{ \sqrt{\vartheta_{tensors}(n, 1, \kappa = \frac{1}{\epsilon_2}, \epsilon_s n^{3/2})}, \frac{\gamma_{min}^4 w_{min}}{4} \right\}$ and $\vartheta_{tensors}$ is the error bound attained in Theorem 2.6 of [25].*

*Proof.* Let $\epsilon' = \vartheta_{8.1}$. The entry-wise sampling error is $\epsilon_s \leq 3 \log n / \sqrt{N}$. Hence, the rank-2 decomposition for $T^{(abc)}$ is $n^{3/2}\epsilon_s$ close in Frobenius norm. We use the algorithm given in [25] to find a rank-2 decomposition of $T^{(abc)}$ that is $O(\epsilon_s)$ close in Frobenius norm. Further, the rank-1 term $u^{(a)} \otimes u^{(b)} \otimes u^{(c)}$ is $\epsilon'^2$-close to $w_1 c_3(\phi_1) x^{(a)} \otimes x^{(b)} \otimes x^{(c)}$. Let us renormalize so that $\|u^{(a)}\| = \|u^{(b)}\| = \|u^{(c)}\| \geq w_{\min}^{1/3} \gamma_{\min}$.

Applying Lemma 13.1, we see that $u^{(a)} = \alpha_a x^{(a)} + z_1^{(a)}$ where $\|z_1^{(a)}\| \leq \epsilon'$, and similarly $v^{(a)} = \beta_a y^{(a)} + z_2^{(a)}$ where $\|z_2\| \leq \epsilon'$. Further $w_{\min}^{1/3} \gamma_{\min} \phi_1 / 4 \leq \alpha_a \leq 1/\gamma_{\min}$. Further

$$\sigma_2\left(\alpha_a x^{(a)}; \beta_a y^{(a)}\right) \geq \min\{\alpha_a, \beta_a\} \sigma_2(M_a) \geq \frac{w_{\min}^{1/3} \gamma_{\min} \phi_1}{4} \sigma_2(M_a).$$

Hence, $\sigma_2(M_a') \geq w_{\min}^{1/3} \gamma_{\min} \phi_1 \sigma_2(M_a)/2 - 2\epsilon'$, as required. The same proof also works for $M_b'$, $M_c'$.
□

Instead of using the enumeration algorithm of [25], the simultaneous eigen-decomposition algorithms in [23] and [26] can also be used. The only difference is that the "full-rank conditions" involving the $M_a, M_b, M_c$ are checked in advance, using the empirical second moment. Note that TENSOR-DECOMP only relies on elements that have a non-negligible chance of appearing in the first position $L$: this can lead to large speedup for constant $\phi_1, \phi_2 < 1$ by restricting to a much smaller tensor.

Lemma 3.2 captures how Algorithm 1 (steps 3 (a - e.i)) performs the first stage using Algorithm 4 and recovers the weights $w_1, w_2$ and $x, y$ when the factor matrices $M_a, M_b, M_c$ are well-conditioned.

In the proof we show that in this case, for one of the $O(\log n)$ random partitions, Lemma 8.1 succeeds and recovers vectors $u^{(a)}, v^{(a)}$ which are essentially parallel to $x^{(a)}$ and $y^{(a)}$ respectively (similarly for $u^{(b)}, u^{(c)}, v^{(b)}, v^{(c)}$). Sorting the entries of $u^{(a)}$ would give the relative ordering among those in $S_a$ of the top few elements of $\pi_1$. However, to figure out all the top-$k$ elements, we need to figure out the correct scaling of $u^{(a)}, u^{(b)}, u^{(c)}$ to obtain $x^{(a)}$. This is done by setting up a linear system.

Now we present the complete proof of the lemmas.

## 8.1 Proof of Lemma 3.2: the Full Rank Case

If such a partition $S_a^*, S_b^*, S_c^*$ exists such that $\sigma_2(M_a) \geq \epsilon_2'$, then there exists a 2-by-2 submatrix of $M_a$ corresponding to elements $e_{i_1}, e_{j_1}$ which has $\sigma_2(\cdot) \geq \epsilon_2'$. Similarly there exists such pairs of elements $e_{i_2}, e_{j_2}$ and $e_{i_3}, e_{j_3}$ in $S_b$ and $S_c$ respectively. But with constant probability the random partition $S_a, S_b, S_c$ has $e_{i_1}, e_{j_1} \in S_a$, $e_{i_2}, e_{j_2} \in S_b$, $e_{i_3}, e_{j_3} \in S_c$ respectively. Hence in the $O(\log n)$ iterations, at least one iteration will produce sets $S_a, S_b, S_c$ such that $\sigma_2(M_a), \sigma_2(M_b), \sigma_2(M_c) \geq \epsilon_2'$ with high probability. Further, Lemma 8.1 also ensures that $\sigma_2(M_a'), \sigma_2(M_b'), \sigma_2(M_c') \geq \epsilon_2$.

Lemma 8.1 recovers vectors $u^{(a)}, v^{(a)}$ which are essentially parallel to $x^{(a)}$ and $y^{(a)}$ respectively (similarly for $u^{(b)}, u^{(c)}, v^{(b)}, v^{(c)}$). While sorting the entries of $u^{(a)}$ would give the relative ordering among those in $S_a$ of the top few elements of $\pi_1$, we need to figure out the correct scaling of $u^{(a)}, u^{(b)}, u^{(c)}$ to recover the top few elements of $\pi_1$.

From Lemma 8.1, we can express

$$w_1 x^{(a)} = \alpha_a' u^{(a)} + z_1^{(a)} \text{ where } z_1^{(a)} \perp u^{(a)} \text{where } \|z_1^{(a)}\| \leq \vartheta_{8.1}(n, \epsilon_s, \epsilon_2').$$

Similarly $w_2 y^{(a)} = \beta_a' v^{(a)} + z_2^{(a)}$, where $\|z_2^{(a)}\| \leq \vartheta_{8.1}$. If $\epsilon_s$ is the sampling error for each entry in $p^{(a)}$, we have

$$\|w_1 x^{(a)} + w_2 x^{(b)} - p^{(a)}\| < \sqrt{n}\epsilon_s \tag{3}$$

$$\|\alpha_a' u^{(a)} + \beta v^{(a)} - p^{(a)}\| < \sqrt{n}\epsilon_s + \frac{1}{2} w_{\min}^{1/3} \phi_1 \gamma_{\min} \vartheta_{8.1} \tag{4}$$

Eq (4) allows us to define a set of linear equations with unknowns $\alpha_a', \beta_a'$, constraint matrix given by $M_a' = (u^{(a)}; v^{(a)})$. Hence, the error in the values of $\alpha_a', \beta_a'$ is bounded by the condition number of the system and the error in the values i.e.

$$\epsilon_\alpha \leq \kappa(M_a').w_{\min}^{1/3} \gamma_{\min} \vartheta_{8.1} \leq \left(\frac{1}{4} w_{\min}^{1/3} \phi_{\min} \gamma_{\min} \epsilon_2' - \vartheta_{8.1}\right)^{-1} \cdot \frac{\phi_{\min}}{2} w_{\min}^{1/3} \gamma_{\min} \vartheta_{8.1}.$$

The same holds for $\alpha_b, \alpha_c, \beta_b, \beta_c$.

**Algorithm 5** REMOVE-COMMON-PREFIX, **Input:** a set $\mathcal{S}$ of $N$ samples from $w_1 \mathcal{M}(\phi, \pi_1) \oplus w_2 \mathcal{M}(\phi, \pi_2)$, $\epsilon$.

---

1. Initialize $I \leftarrow \emptyset$, $S = [n]$.
2. for $t = 1$ to $n$,
   (a) For each element $x \in [n] \setminus I$, estimate $\hat{p}_{x,1} = Pr(x \text{ goes to position } t)$.
   (b) Let $x_t = \arg\max_{x \in [n] \setminus I} \hat{p}_{x,1}$.
   (c) If $|\hat{p}_{x,1} - \frac{1}{Z_{n-t+1}}| > \vartheta(\epsilon)$, return $I$ and QUIT.
   (d) Else $I \leftarrow I \cup x_t$
3. Output $I$.

---

However, we also know that $\|x^{(a)}\|_1 + \|x^{(b)}\|_1 + \|x^{(c)}\|_1 = 1$. Hence,

$$|\|\alpha_a u^{(a)}\|_1 + \|\alpha_b u^{(b)}\|_1 + \|\alpha_c u^{(c)}\|_1 - w_1| \le \epsilon \le 3\sqrt{n}(\epsilon_\alpha + \vartheta_{8.1}).$$

Thus, $\widehat{w}_1, \widehat{w}_2$ are within $\epsilon$ of $w_1, w_2$. Hence, we can recover vectors $x$ by concatenating $\frac{\alpha_a}{w_1} u^{(a)}, \frac{\alpha_b}{w_1} u^{(b)}, \frac{\alpha_c}{w_1} u^{(c)}$ (similarly $y$). Since we have $\vartheta_{8.1} < \phi_1(1-\phi)/w_{\min}$, it is easy to verify that by sorting the entries and taking the ratio of the top two entries, $\widehat{\phi}_1$ estimates $\phi_1$ up to error $\frac{2\vartheta_{8.1}\phi_1(1-\phi_1)}{w_{\min}}$ (similarly $\phi_2$). Finally, since we recovered $x$ up to error $\epsilon'' = \frac{2\vartheta_{8.1}}{w_{\min}}$, we recovered the top $m$ elements of $\pi_1$ where $m \le \log_{\phi_1}(2\vartheta_{8.1}(1-\phi_1)/w_{\min})$.

# 9 Degenerate Case

While we know that we succeed when $M_a, M_b, M_c$ have non-negligible minimum singular value for one of the the $O(\log n)$ random partitions, we will now understand when this does not happen.

Recollect that $L = L_{\sqrt{\epsilon}} = \{e_i : P_i \ge \sqrt{\epsilon}\}$. For every $\ell \in \mathbb{Z}$, let $B_\ell = \{e_i \in L : \pi_1^{-1}(i) - \pi_2^{-1}(i) = \ell\}$. Further let $\ell^*$ be the majority bucket for the elements in $L$. We call a mixture model $w_1 \mathcal{M}(\phi_1, \pi_1) \oplus w_2 \mathcal{M}(\phi_2, \pi_2)$ as degenerate if the parameters of the two Mallows models are equal, and except for at most 2 elements, all the elements in $L$ fall into the majority bucket. In other words, $|\ell^*| \ge |L| - 2$.

We first show that if the tensor method fails, then the parameters of the two models $\phi_1$ and $\phi_2$ are essentially the same. Further, we show how the algorithm finds this parameter as well.

**Lemma 9.1** (Equal parameters). *In the notation of the Algorithm 1, for any $\epsilon' > 0$, suppose $\sigma_2(M_a') < \epsilon_2 \le \vartheta_{9.1}(n, \epsilon', w_{\min}, \phi_1, \phi_2)$ (or $M_b', M_c'$), then with high probability $(1 - 1/n^3)$, we have that $|\phi_1 - \phi_2| \le \epsilon'$ and further Algorithm 9 (ESTIMATE-PHI) finds $|\widehat{\phi} - \phi_1| \le \epsilon'/2$. The number of samples needed $N > poly(n, \frac{1}{\epsilon'})$.*

This lemma is proven algorithmically. We first show that Algorithm 9 finds a good estimate $\widehat{\phi}$ of $\phi_1$. However, by the same argument $\widehat{\phi}$ will also be a good estimate of $\phi_2$! Since $\widehat{\phi}$ will be $\epsilon'/2$-close to both $\phi_1$ and $\phi_2$, this will imply that $|\phi_1 - \phi_2| \le \epsilon'$ ! We prove this formally in the next section. But first, we first characterize when the tensor $T^{(abc)}$ does not have a unique decomposition — this characterization of uniqueness of rank-2 tensors will be crucial in establishing that $\phi_1 \approx \phi_2$.

## 9.1 Characterizing the Rank and Uniqueness of tensor $T^{(abc)}$ based on $M_a, M_b, M_c$

To establish Lemma 9.1, we need the following simple lemma, which establishes that the conditioning of the matrices output by the Algorithm TensorDecomp is related to the conditioning of the parameter matrices $M_a, M_b, M_c$.

**Lemma 9.2** (Rank-2 components). *Suppose we have sets of vectors $(g_i, h_i, g_i', h_i')_{i=1,2,3}$ with length at most one $(\|\cdot\|_2 \le 1)$ such that*

$$T = g_1 \otimes g_2 \otimes g_3 + h_1 \otimes h_2 \otimes h_3 \text{ and } \|T - g_1' \otimes g_2' \otimes g_3' + h_1' \otimes h_2' \otimes h_3'\| \le \epsilon_s$$

such that matrices have minimum singular value $\sigma_2(g_1; h_1), \sigma_2(g_2; h_2) \geq \lambda$ and $\|g_3\|, \|h_3\| \geq \gamma_{min}$, then we have that for matrices $M_1' = (g_1'; h_1'), M_2' = (g_2'; h_2')$

$$\sigma_2(M_1') \geq \frac{\lambda^2 \gamma_{min}}{4n} - \epsilon_s \text{ and } \sigma_2(M_1') \geq \frac{\lambda^2 \gamma_{min}}{4n} - \epsilon_s.$$

*Proof.* Let matrices $M_1 = (g_1; h_1), M_2 = (g_2; h_2)$. For a unit vector $w$ (of appropriate dimension) let

$$M_w = T(\cdot, \cdot, w) = \langle w, g_3 \rangle g_1 \otimes g_2 + \langle w, h_3 \rangle h_1 \otimes h_2$$
$$= M_1 D_w M_2^T \text{ where } D_w = \begin{pmatrix} \langle w, g_3 \rangle & 0 \\ 0 & \langle w, h_3 \rangle \end{pmatrix}.$$

Besides, since $w$ is a random gaussian unit vector, $\mathbf{Pr}|\langle w, g_3 \rangle| \geq \|g_3\|/4\sqrt{n}$ with probability $> 1/2$. Hence, using there exists a unit vector $w$ such that $\min\{|\langle w, g_3 \rangle|, |\langle w, h_3 \rangle|\} \geq \gamma_{min}/(4\sqrt{n})$. Hence,

$$\sigma_2(M_w) \geq \frac{\lambda^2 \gamma_{min}}{4\sqrt{n}}.$$

However, $\|M_w - M_1' D_w' (M_2')^T\|_F \leq \epsilon_s$ where $D_w' = \begin{pmatrix} \langle w, g_3' \rangle & 0 \\ 0 & \langle w, h_3' \rangle \end{pmatrix}.$

Hence, $\sigma_2\left(M_1' D_w' (M_2')^T\right) \geq \sigma_2(M_w) - \epsilon_s$.
Combining this with the fact that $\sigma_2\left(M_1' D_w' (M_2')^T\right) \leq \sigma_2(M_1')\sigma_1(D_w')\sigma_1(M_2')$ gives us the claimed bound. $\qquad \square$

This immediately implies the following lemma in the contrapositive.

**Lemma 9.3** (Rank-1 components). *Suppose $\sigma_2(M_a') < \epsilon$ and $\sigma_2(M_b') < \epsilon$, then two of the matrices $M_a, M_b, Mc$ have $\sigma_2(\cdot) < \sqrt{\frac{8\epsilon n}{\gamma_{min}}}$, when the number of samples $N > poly(n, 1/\epsilon)$.*

## 9.2 Equal Scaling Parameters

The following simple properties of our random partition will be crucial for our algorithm.

**Lemma 9.4.** *The random partition of $[m]$ into $A, B, C$ satisfies with high probability (at least $1 - exp\left(-\frac{1}{C_{9.4}} \cdot m\right)$):*

1. *$|A|, |B|, |C| \geq m/6$*

2. *There are many consecutive numbers in each of the three sets $A, B, C$ i.e.*

$$|\{i \in A \text{ and } i + 1 \in A\}| \geq m/100.$$

*Proof.* The claimed bounds follow by a simple application of Chernoff Bounds, since each element is chosen in $A$ with probability $1/3$ independently at random. The second part follows by considering the $m/2$ disjoint consecutive pairs of elements, and observing that each pair fall entirely into $A$ with probability $1/9$. $\qquad \square$

**Lemma 9.5.** *Consider a set of indices $S \subseteq [n]$ and let $p_S$ be the true probability vector $p$ of a single Mallows model $\mathcal{M}(\pi, \phi)$ restricted to subset $S$. Suppose the empirical vector $\|\hat{p}_S - p_S\|_\infty < \epsilon_1$, and there exists consecutive elements of $\pi$ in $S$ i.e. $\exists i$ such that $\pi(i), \pi(i+1) \in S$, with $p(\pi(i+1)) \geq \sqrt{\epsilon_1}$. Then, if we arrange the entries of $p_S$ in decreasing order as $r_1, r_2, \ldots, r_{|S|}$ we have that*

$$\hat{\phi} = \max_{i:r_{i+1} \geq \sqrt{\epsilon_1}} \frac{r_{i+1}}{r_i} \text{ satisfies } |\hat{\phi} - \phi| < 2\sqrt{\epsilon_1}.$$

*Proof.* By the properties of the Mallows model, the ratio of any two probabilities is a power of $\phi$ i.e. $\frac{p_{\ell_2}}{p_{\ell_1}} = \phi^{\pi^{-1}(\ell_2) - \pi^{-1}(\ell_1)}$. If $p(\pi(i+1)) \geq \sqrt{\epsilon_1}$, we have that

$$\frac{\hat{p}(\pi(i+1))}{\hat{p}(\pi(i))} \leq \frac{\phi \cdot p(\pi(i)) + \epsilon_1}{p(\pi(i)) - \epsilon_1}$$

$$\leq \phi + \frac{\phi\left(p(\pi(i)) - \hat{p}(\pi(i)) + \epsilon_1\right)}{\hat{p}(\phi(i))} \leq \phi + \epsilon_1 \frac{(1+\phi)}{\hat{p}(\phi(i))} \qquad \leq \phi + 2\sqrt{\epsilon_1}$$

The same proof holds for the lower bound. $\qquad\square$

We now proceed to showing that the scaling parameters are equal algorithmically.

*Proof of Lemma 9.1.* We now proceed to prove that $\phi_1 \approx \phi_2$. We note that $\|T^{(abc)}\|_F \leq 1$ since the entries of $T^{(abc)}$ correspond to probabilities, and for any vector $z$, $\|z\|_2 \leq \|z\|_1$. This implies that all the vectors in the decomposition can be assumed to have $\ell_2$ norm at most 1, without loss of generality. We can first conclude that at least one of the three matrices $M_a, M_b, M_c$ has $\sigma_2(\cdot) < \sqrt{\frac{8n\epsilon_2}{\gamma_{\min}w_{\min}}}$. Otherwise, we get a contradiction by applying Lemma 9.2 (contrapositive) to $M'_a, M'_b$ and $M'_a, M'_c$. Now, we will show how the algorithm gives an accurate estimate $\widehat{\phi}$ of $\phi_1$. However the exact argument applied to $\phi_2$ will show that $\widehat{\phi}$ is also a good estimate for $\phi_2$, implying that $\phi_1 \approx \phi_2$.

We have two cases depending on whether one of $\sigma_2(M'_b)$ and $\sigma_2(M'_c)$ are non-negligible or not.

**Case 1:** $\sigma_2(M'_b) \geq (\epsilon_2^{1/4}\frac{8n}{\gamma_{\min}w_{\min}})^{3/4})$ **and** $\sigma_2(M'_c) \geq (\epsilon_2^{1/4}\frac{8n}{\gamma_{\min}w_{\min}})^{3/4})$:
Applying Lemma 9.2, we conclude that $\sigma_2(M_b) \geq \epsilon_2^{1/2}(8n/\gamma_{\min}w_{\min})^{1/2}$ and $\sigma_2(M_c) \geq \epsilon_2^{1/2}(\frac{8n}{\gamma_{\min}w_{\min}})^{1/2}$. However one of the matrices $M_a, M_b, M_c$ has small $\sigma_2$ value. Hence

$$\sigma_2(M_a) < \epsilon_2^{1/2}\left(\frac{8n}{\gamma_{\min}w_{\min}}\right)^{1/2} = \epsilon'_2 \text{ (say)}.$$

Let $y^{(a)} = \alpha x^{(a)} + y^{\perp}$ where $y^{\perp} \perp x^{(a)}$. Then $\|y^{\perp}\| \leq \epsilon'_2$ and $\alpha \geq \frac{(\|y^{(a)}\| - \epsilon'_2)}{\|x^{(a)}\|} \geq \gamma_{\min}/2$. Further, $p^{(a)} = (w_1 + w_2\alpha)x^{(a)} + w_2 y^{\perp}$. Hence,

$$x^{(a)} = \beta p^{(a)} - w_2\beta y^{\perp}, \quad \text{where } 0 \leq \beta < \frac{2}{\gamma_{\min}}.$$

Since the sampling error is $\epsilon_s$, we have

$$x^{(a)} = \beta\hat{p}^{(a)} + \beta(p^{(a)} - \hat{p}^{(a)}) - w_2\beta y^{\perp}$$

$$= \beta\hat{p}^{(a)} + z \text{ where } \|z\|_{\infty} \leq \beta(\epsilon_s + \epsilon'_2) \leq \frac{4\epsilon_2}{\gamma_{\min}} = \epsilon_3$$

Consider the first $m = C_{9.4}\log n$ elements $F$ of $\pi_1$.

$$\forall i \in F, x_i \geq \frac{\phi_1^{C_{9.4}\log n}}{1 - \phi_1} \geq \frac{n^{C_{9.4}\log(1/\phi_1)}}{1 - \phi_1}$$

$$\geq \sqrt{\epsilon_3} \text{ due to our choice of error parameters}$$

Applying Lemma 9.4, $\Omega(\log n)$ consecutive elements of $\pi_1$ occur in $S_a$. Hence applying Lemma 9.5, we see that the estimate $\widehat{\phi}$ output by the algorithm satisfies $|\widehat{\phi} - \phi_1| \leq 2\sqrt{\epsilon_3} = \frac{16\epsilon_2^{1/4}n^{1/4}}{\gamma_{\min}^{3/4}w_{\min}^{1/4}}$, as required.

**Case 2:** $\sigma_2(M'_b) < (\epsilon_2^{1/4}\frac{8n}{\gamma_{\min}w_{\min}})^{3/4})$:
We also know that $\sigma_2(M'_a) < \epsilon_2$. Applying Lemma 9.3, we see that two of the three matrices $M_a, M_b, M_c$ have $\sigma_2(\cdot)$ being negligible i.e.

$$\sigma_2(\cdot) < \epsilon_2^{1/4}\left(\frac{8n}{\gamma_{\min}w_{\min}}\right)^{7/8}.$$

**Algorithm 6** HANDLE-DEGENERATE, **Input:** a set $\mathcal{S}$ of $N$ samples from $w_1\mathcal{M}(\phi_1, \pi_1) \oplus w_2\mathcal{M}(\phi_2, \pi_2), \widehat{\phi}$.

---

1. $\pi^{pfx} \leftarrow$ REMOVE-COMMON-PREFIX($\mathcal{S}$). Let $\pi_1^{rem} = \pi_1 \setminus \pi^{pfx}, \pi_2^{rem} = \pi_2 \setminus \pi^{pfx}$.
2. If $|\pi^{pfx}| = n$, then output IDENTICAL MALLOWS MODELS and parameters $\widehat{\phi}$ and $\pi^{pfx}$.
3. Let $\mathcal{M}'$ be the Mallows mixture obtained by adding three artificial elements $e_1^*, e_2^*, e_3^*$ to the front.
4. Run steps (1-3) of Algorithm 1 on $\mathcal{M}'$. If SUCCESS, output $\widehat{w}_1, \widehat{w}_2, \pi_1, \pi_2, \widehat{\phi}$.
5. If FAIL, let $\widehat{P}(i), \widehat{P}(i,j)$ be the estimates of $P(i), P(i,j)$ when samples according to $\mathcal{M}'$.
6. Divide elements in $L_{\sqrt{\epsilon}}$ into $R \leq \frac{\log(\epsilon Z_n(\widehat{\phi}))}{2\log(\widehat{\phi})}$ disjoint sets

$$I_r = \left\{ i : \widehat{P}(i) \in \left[ \frac{\widehat{\phi}^r}{Z_n(\widehat{\phi})} - \epsilon, \frac{\widehat{\phi}^r}{Z_n(\widehat{\phi})} + \epsilon \right] \right\}.$$

7. If $|I_r| = 1$ set $\pi_1^{rem}(i)$ to be the only element in $I_r$.
8. Let $I_{bad}$ be the remaining elements in the sets $I_1 \cup I_2 \ldots I_R$ along with $L_{\sqrt{\epsilon}} \setminus \bigcup_r I_r$. If $|I_{bad}| > 4$ or $|I_{bad}| < 2$, output FAIL.
9. Let $S_a, S_b$ is any partition of $I_1 \cup I_2 \cup I_R \setminus I_{bad}$.
   Find $i_1, j_1 \in I_{bad}$ such that $M = \left(\widehat{P}_{ij}\right)_{i \in S_a \cup \{i_1\}, S_b \cup \{j_1\}}$ has $\sigma_2(M) \geq \sqrt{\epsilon}n$.
10. For $i \in I_{bad} \setminus \{i_1, j_1\}$ and $i \in I_r$, set $\pi_1^{rem}(r) = \pi_2^{rem}(r) = i$.
    Set $\pi_1^{rem}(1) = i_1, \pi_2^{rem}(1) = j_1$, and $\pi_1^{rem}(k) = j_1, \pi_2^{rem}(k) = i_1$ where $k \leq R$ is unfilled position.
11. Output $\pi_1 = \pi^{pfx} \circ \pi_1^{rem}, \pi_2 = \pi^{pfx} \circ \pi_2^{pfx}, \widehat{\phi}$.
    Output $\widehat{w}_1, \widehat{w}_2 = 1 - \widehat{w}_1$, by solving for $\widehat{w}_1$ from $\widehat{P}(i) = \widehat{w}_1\pi_1^{-1}(i) + (1 - \widehat{w}_1)\pi_2^{-1}(i)$.

---

Using the same argument as in the previous case, we see that the estimates given by two of the three partitions $S_a, S_b, S_c$ is $2\sqrt{\epsilon_3}$ close to the $\phi_1$. Hence the median value $\widehat{\phi}$ of these estimates is also as close.

As stated before, applying the same argument for $\phi_2$ (and $\pi_2$), we see that $\widehat{\phi}$ is $2\sqrt{\epsilon_3}$ close to $\phi_2$ as well. Hence, $\phi_1$ is $4\sqrt{\epsilon_3}$ close to $\phi_1$.

$\square$

## 9.3 Establishing Degeneracy

Next, we establish that if none of the $O(\log n)$ rounds were successful, then the two central permutations (restricted to the top $O(\log_{1/\phi_{\min}} n)$ positions) are essentially the same shifted by at most a couple of elements.

**Lemma 9.6.** *Consider the large elements $L_{\sqrt{\epsilon}}$. Suppose $|B_{\ell*}| \geq |L_{\sqrt{\epsilon}}| - 3$, then the one of the $O(\log n)$ rounds of the Tensor Algorithm succeeds with high probability.*

*Proof.* We have two cases depending on whether $\ell^* \leq \log(\epsilon)/\log(\phi)$ or not.

Suppose $|\ell^*| \leq \log(\epsilon(1-\phi))/\log(\phi)$. Let $i, j, k$ be the indices of elements in $L_{\sqrt{\epsilon}}$ that are not in $B_{\ell*}$. With constant probability the random partition $S_a, S_b, S_c$ puts these three elements in different partitions. In that case, by applying Lemma 9.11 we see that $\sigma_2(M_a), \sigma_2(M_b), \sigma_2(M_c) \geq \epsilon^2(1-\phi)^2$. Hence, Lemma 3.2 would have succeeded with high probability.

Suppose $|\ell^*| > \log(\epsilon(1-\phi))/\log(\phi)$. Assume without loss of generality that $\ell* \geq 0$. Consider the first three elements of $\pi_2$. They can not belong to $B_{\ell*}$ since $\ell* > 3$. Hence, by pairing each of these elements with some three elements of $B_{\ell*}$, and repeating the previous argument we get that $\sigma_2(M_a), \sigma_2(M_b), \sigma_2(M_c) \geq \epsilon^2(1-\phi)^2$ in one of the iterations w.h.p. Hence, Lemma 3.2 would have succeeded with high probability. $\square$

Hence we now have two kinds of degenerate cases to deal with. The next two lemmas show how such cases are handled.

**Lemma 9.7** (Staggered degenerate case). *Suppose $\phi = \phi_1 = \phi_2$, and at most two of the top elements $L_{\sqrt{\epsilon}}$ are not in bucket $B_{\ell^*}$ i.e. $B_{\ell^*} \geq L_{\sqrt{\epsilon}} - 2$ with $\ell^* \neq 0$. Then, for any $\epsilon > 0$, given $N > poly(n, \phi, \epsilon, w_{min})$ samples, step (3-4) of Algorithm* HANDLE-DEGENERATE *finds finds $\widehat{w}_1, \widehat{w}_2$ of $w_1, w_2$ up to $\epsilon$ accuracy and the top $m$ elements of $\pi_1, \pi_2$ respectively where $m = \frac{\log Z_n(\epsilon)}{2 \log \phi}$.*

*Proof.* Since $\phi_1 = \phi_2$, we can use Lemma 6.3, we can sample from a Mallows mixture where we add one new element $e_3^*$ to the front of both permutations $\pi_1, \pi_2$. Doing this two more times we can sample from a Mallows mixture where we add $e_1^*, e_2^*, e_3^*$ to the front of both permutations. Let these new concatenated permutations be $\pi_1^*, \pi_2^*$. Since the majority bucket corresponds to $\ell^* \neq 0$, we have at least three pairs of elements which satisfy Lemma 9.11, we see that w.h.p. in one of the $O(\log n)$ iterations, the partitions $S_a, S_b, S_c$ have $\sigma_2(\cdot) \geq (\gamma_{min}\phi^6)^2(1 - \phi)^3$.

Hence, by using the Tensor algorithm with guarantees from Lemma 3.2, and using Algorithm RECOVER-REST, we get the full rankings as required (using Lemma 10.2). $\quad\square$

**Lemma 9.8** (Aligned Degenerate case). *Suppose $\phi = \phi_1 = \phi_2$, and at most two of the top elements $L_{\sqrt{\epsilon}}$ are not in bucket $B_0$ i.e. $|B_0| \geq |L_{\sqrt{\epsilon}}| - 2$. For any $\epsilon > 0$, given $N = O(\frac{n^2 \log n}{\epsilon^8 w_{min}^2 (1-\phi)^4})$ samples, steps (5-10) of Algorithm 6 (*HANDLE-DEGENERATE*) finds estimates $\widehat{w}_1, \widehat{w}_2$ up to $\epsilon$ accuracy and prefixes $\pi_1', \pi_2'$ of $\pi_1, \pi_2$ respectively that contain at least the top $m$ elements where $m = \frac{\log Z_n(\epsilon)}{2 \log \phi}$.*

*Proof.* The first position differs because of step(1-2) of Algorithm 6. Without loss of generality $\pi_1 = \pi_1^{pfx}$ and $\pi_2 = \pi_2^{pfx}$. $B_0 \geq m - 2$, hence $|B_0| = m - 2$. Let $e_{i_1}, e_{j_1}$ be the other two elements in $L_{\sqrt{\epsilon}}$.

For elements $e_i \in B_0, \pi_1^{-1}(i) = \pi_2^{-1}(i)$. The sampling error in the entries of $\widehat{P}$ is at most $\epsilon_s = \epsilon^4 w_{min}(1 - \phi)^2/n$. Hence, they fall into the set $I_{\pi_1^{-1}(i)}$. Therefore, there can be at most four sets with at most four elements between them that constitute $I_{bad}$.

Consider $M = (\widehat{P}_{ij})_{i \in S_a \cup \{i_1\}, j \in S_b \cup \{j_1\}}$. Also let $M_a = (x^{(a)}; y^{(a)})$ and $M_b = (x^{(a)}; y^{(a)})$ applied to $\mathcal{M}'$. By Lemma 9.11, we see that $\sigma_2(M_a), \sigma_2(M_b) \geq \epsilon(1 - \phi)$. Further,

$$\|M - M_a \begin{pmatrix} w_1 & 0 \\ 0 & w_2 \end{pmatrix} M_b^T\|_F \leq \epsilon_s n.$$

Hence, $\sigma_2(M) \geq \epsilon^2(1 - \phi)^2 w_{min}$. If $i_1, j_1$ do not belong to the two different partitions $S_a, S_b$, it is easy to see that $\sigma_2(M) \leq \sqrt{\epsilon_s} > \epsilon^2 w_{min}(1 - \phi)^2$. Hence, we identify the two irregular elements that are not in bucket $B_0$, and use this to figure out the rest of the permutations. $\quad\square$

Finally, the following lemma shows how the degenerate cases are handled.

**Lemma 9.9.** *For $0 < \epsilon$, given $\phi_1, \phi_2$ with $|\phi_1 - \phi_2| \leq \epsilon_1 = \vartheta_{9.9}(n, \phi, \epsilon, w_{min})$, such that at most two elements of $L_{\sqrt{\epsilon}}$ are not in the bucket $B_{\ell^*}$, then Algorithm* HANDLE-DEGENERATE *finds w.h.p. estimates $\widehat{w}_1, \widehat{w}_2$ of $w_1, w_2$ up to $\epsilon$ accuracy, and recovers $\pi_1, \pi_2$.*

*Proof.* We can just consider the case $\widehat{\phi} = \widehat{\phi}_1 = \phi_2$ using Lemma 2.6 as long as $\epsilon_1 < \frac{\phi}{n^2 N(n, \phi, \epsilon)^2}$, where $N$ is the number of samples used by Lemma 9.8 and Lemma 9.7 to recover the rest of the permutations and parameters up to error $\epsilon$. This is because the simulation oracle does not fail on any of the samples w.h.p, by a simple union bound.

If the two permutations do not differ at all, then by Lemma 10.6, Algorithm 5 returns the whole permutation $\pi_1 = \pi_2$. Further, any set of weights can be used since both are identical models ($\phi_1 = \phi_2 = \phi$).

Let $m = L_{\sqrt{\epsilon}}$. In the remaining mixture $\mathcal{M}'$, the first position of the two permutations differ: hence, $B_{\ell^*} < m$. Further, we know that $B_{\ell^*} \geq m - 2$.

We have two cases, depending on whether the majority bucket $B_{\ell^*}$ corresponds to $\ell^* = 0$ or $\ell^* \neq 0$. In the first case, Lemma 9.7 shows that we find the permutations $\pi_1, \pi_2$ and parameters up to

accuracy $\epsilon$. If this FAILS, we are in the case $\ell^* = 0$, and hence Lemma 9.8 shows that we find the permutations $\pi_1, \pi_2$ and parameters up to accuracy $\epsilon$. $\qquad\square$

## 9.4 Auxiliary Lemmas for Degenerate Case

**Lemma 9.10.** *For any Mallows model with parameters $\phi_1, \phi_2$ has*

$$\gamma_{min} = \min_{\tau \in \{a,b,c\}} \min \left\{ \|x^{(\tau)}\|, \|y^{(\tau)}\| \right\} \geq \min \left\{ \phi_1^{2C \log n}(1 - \phi_1), \phi_2^{2C \log n}(1 - \phi_2) \right\} \text{ with probability } 1 - n^C$$

*Proof.* Consider a partition $A$, and the top $m \geq 2C \log n$ elements according to $\pi$. The probability that none of the them belong to $A$ is at most $1/n^C$. This easily gives the required conclusion. $\qquad\square$

**Lemma 9.11.** *When $\phi_1 = \phi_2 = \phi$, if two large elements $e_i, e_j \in L_{\sqrt{\epsilon}}$ belonging to different buckets $B_{\ell_1}$ and $B_{\ell_2}$ respectively with $\max \{|\ell_1|, |\ell_2|\} \leq \frac{\log(\epsilon)}{\log(\phi)}$. Suppose further that these elements are in the partition $S_a$. Then the corresponding matrix $M_a$ has $\sigma_2(M_a) \geq \epsilon^2(1 - \phi)$ when $\phi_1 = \phi_2 = \phi$.*

*Proof.* Consider the submatrix

$$M = \begin{pmatrix} x_i & y_i \\ x_j & y_j \end{pmatrix} = x_i \begin{pmatrix} 1 & \phi^{\ell_1} \\ \phi^{\pi_1^{-1}(i) - \pi_1^{-1}(j)} & \phi^{\pi_1^{-1}(i) - \pi_1^{-1}(j)} \cdot \phi^{\ell_2} \end{pmatrix}.$$

Using a simple determinant bound, it is easy to see that

$$\sigma_1(M)\sigma_2(M) \geq \max \{x_i, y_i\} \max \{x_j, y_j\} \cdot (\phi^{|\ell_1|} - \phi^{|\ell_2|}) \geq \max \{x_i, y_i\} \cdot \epsilon \phi^{\min\{|\ell_1|, |\ell_2|\}}(1 - \phi).$$

Since $\sigma_1(M) \leq 4 \max x_i, y_i$, we see that $\sigma_2(M) \geq \frac{\epsilon^2(1 - \phi)}{4}$. $\qquad\square$

# 10 Recovering the complete rankings

Let $f^{(1)}(i \to j)$ be the probability that element $e_i$ goes to position $j$ according to Mallows Model $\mathcal{M}_1$ (and similarly $f^{(2)}(i \to j)$ for model $\mathcal{M}_2$). To find the complete rankings, we measure appropriate statistics to set up a system of linear equations to calculate $f^{(1)}(i \to j)$ and $f^{(2)}(i \to j)$ up to inverse polynomial accuracy. The largest of these values $\{f^{(1)}(i \to j)\}$ corresponds to the position of $e_i$ in the central ranking of $\mathcal{M}(,1)$. To compute these values $\{f^{(r)}(i \to j)\}_{r=1,2}$ we consider statistics of the form "*what is the probability that $e_i$ goes to position $j$ conditioned on $e_{i^*}$ going to the first position?*". This statistic is related to $f^{(1)}(i \to j), f^{(2)}(i \to j)$ for element $e_{i^*}$ that is much closer than $e_i$ to the front of one of the permutations.

**Notation:** Let $f_{\mathcal{M}}(i \to j)$ be the probability that element $e_i$ goes to position $j$ according to Mallows Model $\mathcal{M}$, and let $f^{(r)}(i \to j)$ be the same probability for the Mallows model $\mathcal{M}_r$ ($r \in \{1, 2\}$). Let $f^{(1)}(i \to j|e_{i^*} \to 1)$ be the probability that $e_i$ goes to the $j$th position conditioned on the element $e_{i^*}$ going to the first position according to $\mathcal{M}_1$ (similarly $\mathcal{M}(,2)$). Finally for any Mallows model $\mathcal{M}(\phi, \pi)$, and any element $e_{i^*} \in pi$, let $\mathcal{M}_{-i^*}$ represent the Mallows model on $n - 1$ elements $\mathcal{M}(\phi, \pi - i^*)$.

In the notation defined above, we have that for any elements $e_{i^*}, e_i$ and position $j$, we have

$$\mathbf{Pr}(e_i \to j|e_{i^*} \to 1) = w_1' f^{(1)}(i \to j|e_{i^*} \to 1) + w_2' f^{(2)}(i \to j|e_{i^*} \to 1)$$

$$\text{where } w_1' = \frac{w_1 x_{i*}}{w_1 x_{i^*} + w_2 y_{i^*}}, w_2' = 1 - w_1'$$

However, these statistics are not in terms of the unknown variables $f^{(1)}(i \to j), f^{(2)}(i \to j)$. The following lemma shows that these statistics are *almost* linear equations in the unknowns $f^{(1)}(i \to j), f^{(2)}(i \to j)$ for the $i, j$ pairs that we care about. For threshold $\delta$, let $r_1$ be the smallest number $r$ such that $\delta > \phi_1^{r-1}/Z_n(\phi_1)$. Similarly let $r_2$ be the corresponding number for second Mallows models $\mathcal{M}_2$.

**Algorithm 7** RECOVER-REST, **Input:** a set $\mathcal{S}$ of $N$ samples from $w_1 \mathcal{M}(\phi_1, \pi_1) \oplus w_2 \mathcal{M}(\phi_2, \pi_2)$, $\hat{w}_1, \hat{w}_2, \hat{\phi}_1, \hat{\phi}_2, \hat{\pi}_1, \hat{\pi}_2, \epsilon$.

1. Let $|\hat{\pi}_1| = r_1$, $|\hat{\pi}_2| = r_2$ and let $r_1 \geq r_2$ w.l.o.g. (the other case is the symmetric analog).

2. For any element $e_i$, define $\hat{f}^{(1)}(i \to 1) = \frac{\hat{\phi}_1^{\left(\hat{\pi}_1^{-1}(e_i)-1\right)}}{Z_n(\hat{\phi}_1)}$, and $\hat{f}^{(2)}(i \to 1) = \frac{\hat{\phi}_2^{\left(\hat{\pi}_2^{-1}(e_i)-1\right)}}{Z_n(\hat{\phi}_2)}$.
   If $e_i$ does not appear in $\hat{\pi}_1$ set $\hat{f}^{(1)}(i \to 1) = 0$. Similarly, if $e_i$ does not appear in $\hat{\pi}_2$ set $\hat{f}^{(2)}(i \to 1) = 0$. Define $g(n, \phi) = C \cdot \frac{n^2 \phi^2}{(1-\phi)^2} \log n$, where $C$ is an absolute constant.

3. For each $e_i \in \hat{\pi}_1(1 : r_1/2)$

   (a) If $\hat{f}^{(2)}(i \to 1) < \frac{\min\{\hat{w}_1, \hat{w}_2\}}{16} \frac{\hat{f}^{(1)}(i \to 1)}{n^2 g(n, \hat{\phi}_1)}$

      i. $\hat{\pi}_1 \leftarrow$ LEARN-SINGLE-MALLOW$(\mathcal{S}_{e_i \mapsto 1})$. Here $\mathcal{S}_{e_i \mapsto 1}$ refers to the samples in $\mathcal{S}$ where $e_i$ goes to position 1.

      ii. $\hat{\pi}_2 \leftarrow$ FIND-PI$(\mathcal{S}, \hat{\pi}_1, \hat{w}_1, \hat{w}_2, \hat{\phi}_1, \hat{\phi}_2)$. Output SUCCESS and return $\hat{\pi}_1$ and $\hat{\pi}_2$, $\hat{w}_1$, $\hat{w}_2$, $\hat{\phi}_1$ and $\hat{\phi}_2$.

4. Do similar check for each $e_i \in \hat{\pi}_2(1 : r_2/2)$.

5. Let $e_{i^*}$ be the first element in $\hat{\pi}_1$ such that $|\hat{f}^{(1)}(i^* \to 1) - \hat{f}^{(2)}(i^* \to 1)| > \epsilon$. Define $\hat{w}_1' = \frac{1}{1 + \frac{\hat{w}_2}{\hat{w}_1} \frac{\hat{f}^{(2)}(i^* \to 1)}{\hat{f}^{(1)}(i^* \to 1)}}$ and $\hat{w}_2' = 1 - \hat{w}_1'$.

6. For each $e_i \notin \hat{\pi}_1$ and $j > r_1$

   (a) Estimate $\hat{f}(i \to j) = Pr[e_i \text{ goes to position } j]$ and $\hat{f}(i \to j|e_{i^*} \to 1) = Pr[e_i \text{ goes to position } j|e_{i^*} \mapsto 1]$.

   (b) Solve the system

   $$\hat{f}(i \to j) = \hat{w}_1 \hat{f}^{(1)}(i \to j) + \hat{w}_2 \hat{f}^{(2)}(i \to j) \tag{5}$$

   $$\hat{f}(i \to j|e_{i^*} \to 1) = \hat{w}_1' \hat{f}^{(1)}(i \to j) + \hat{w}_2' \hat{f}^{(2)}(i \to j) \tag{6}$$

7. Form the ranking $\hat{\pi}_1 = \hat{\pi}_1 \circ \pi_1'$ s.t. for each $e_i \notin \hat{\pi}_1$, $pos(e_i) = \arg\max_{j > r_1} \hat{f}^{(1)}(i \to j)$.

8. $\hat{\pi}_2 \leftarrow$ FIND-PI$(\mathcal{S}, \hat{\pi}_1, \hat{w}_1, \hat{w}_2, \hat{\phi}_1, \hat{\phi}_2, \epsilon)$. Output SUCCESS and return $\hat{\pi}_1$ and $\hat{\pi}_2$, $\hat{w}_1$, $\hat{w}_2$, $\hat{\phi}_1$ and $\hat{\phi}_2$.

---

**Algorithm 8** LEARN-SINGLE-MALLOW, **Input:** a set $\mathcal{S}$ of $N$ samples from $\mathcal{M}(\phi, \pi)$.

1. For each element $e_i$, estimate $\hat{f}^{(1)}(i \to j) = Pr[e_i \text{ goes to position } j]$.

2. Output a ranking $\hat{\pi}$ such that for all $e_i$, $pos(e_i) = \arg\max_j \hat{f}^{(1)}(i \to j)$.

---

**Lemma 10.1.** *For any $j > r_1$, any elements $e_{i^*}, e_i$ with $pos_{\pi_1}(i^*) > r_1$, $pos_{\pi_1}(i) > pos_{\pi_1}(i^*)$, we have in the notation defined above that*

$$f^{(1)}(i \to j|e_{i^*} \to 1) = f^{(1)}(i \to j) + \delta' \quad \text{where } |\delta'| \leq \delta n.$$

*The corresponding statement also holds for Mallows model $\mathcal{M}_2$.*

*Proof.* When samples are generated according to Mallows model $\mathcal{M}_1$, we have for these sets of $i, i^*, j$ that the conditional probability $f^{(1)}(i \to j|e_{i^*} \to 1) = f_{\mathcal{M}(\phi_1, \pi_1 - i^*)}(i \to j - 1)$, where the term on the right is a Mallows model over $n - 1$ elements.

$$f^{(1)}(i \to j) = \sum_{i'=1}^{n} \mathbf{Pr}(e_{i'} \to 1) f^{(1)}(i \to j|e_{i^*} \to 1) \leq \sum_{i'=1}^{r_1} \mathbf{Pr}(e_{i'} \to 1) f_{\mathcal{M}(\phi_1, \pi_1 - i')}(i \to j - 1) + \delta$$

$$= f_{\mathcal{M}(\phi_1, \pi_1 - i^*)}(i \to j - 1) \sum_{i'=1}^{r_1} \mathbf{Pr}(e_{i'} \to 1).$$

The last equality is because the probability is independent of $i'$ (since $pos_{\pi_1}(e_i) > pos_{\pi_1}(e_{i^*})$). Hence, it follows easily that

$$f^{(1)}(i \to j|e_{i^*} \to 1)(1 - \delta) \leq f^{(1)}(i \to j) \leq f^{(1)}(i \to j|e_{i^*} \to 1) + \delta.$$

---

**Algorithm 9** ESTIMATE-PHI, **Input:** $\widehat{P}$.

---

    1. Sort $P$ in decreasing order. Return $\min_i \{\frac{P_{i+1}}{P_i}\}$.

---

**Algorithm 10** FIND-PI, **Input:** a set $\mathcal{S}$ of $N$ elements from $\widehat{w}_1 \mathcal{M}\left(\widehat{\phi}_1, \pi_1\right) \oplus \widehat{w}_2 \widehat{\mathcal{M}}\left(\widehat{\phi}_2, \pi_2\right)$, $\hat{\pi}_1$, $\hat{w}_1, \hat{w}_2, \hat{\phi}_1, \hat{\phi}_2$.

---

    1. Compute $\hat{f}^{(1)}(i \to j) = \mathbf{Pr}\left(e_i \text{ goes to position } j | \hat{\pi}_1\right)$ (see Lemma 10.8).

    2. For each element $e_i$, estimate $\hat{f}_{e_i,j} = \mathbf{Pr}\left(e_i \text{ goes to position } j\right)$.

    3. Solve for $\hat{f}^{(1)}(i \to j)$ using the equation $\hat{f}_{e_i,j} = \hat{w_1}\hat{f}^{(1)}(i \to j) + \hat{w_2}\hat{f}^{(2)}(i \to j)$.

    4. Output $\hat{\pi}_2$ such that for each $e_i$, $pos(e_i) = \arg\max_j \hat{f}^{(2)}(i \to j)$.

---

$\square$

Hence, by picking an appropriate element $e_{i^*}$, we can set up a system of linear equations and solves for the quantities $\left\{f^{(1)}(i \to j), f^{(2)}(i \to j)\right\}$. Suppose there exists an element $e_{i^*}$ that occurs in the top few positions in both the permutations, then that element would suffice for our purpose. On the other hand, if we condition on an element $i^*$ which occurs near the top in one permutation but far away in the other permutation, gives us a *single* Mallows model. The sub-routine RECOVER-REST of the main algorithm figures out which of the cases we are in, and succeeds in recovering the entire permutations $\pi_1$ and $\pi_2$ in the case that $w_1 \mathcal{M}(\phi_1, \pi_1) \oplus w_2 \mathcal{M}(\phi_2, \pi_2)$ is non-degenerate (the degenerate cases have been handled separately in the previous section). In such a scenario, from the guarantee of Lemma 3.2 we can assume that we have parameters $\{\widehat{w}_1, \widehat{w}_2, \widehat{\phi}_1, \widehat{\phi}_2\}$ which are within $\epsilon \leq \epsilon_0$ of the true parameters. For the rest of this section we will assume that RECOVER-REST and every sub-routine it uses has access to samples from $\widehat{w}_1 \mathcal{M}\left(\widehat{\phi}_1, \pi_1\right) \oplus \widehat{w}_2 \widehat{\mathcal{M}}\left(\widehat{\phi}_2, \pi_2\right)$. This is w.l.o.g. due to Lemma 2.6.

The rankings $\widehat{\pi}_1$ and $\widehat{\pi}_2$ are obtained from INFER-TOP-K. Define $\gamma = \frac{(1-\phi_{\max})^2}{4n\phi_{\max}}$. By our choice of $\epsilon_0$, rankings $|\hat{\pi}_1| = r_1 \geq \log_{1/\phi_1}\left(\frac{n^{10} Z_n(\phi_1)}{w_{\min}^2 \gamma^2}\right)$ and $|\hat{\pi}_2| = r_2 \geq \log_{1/\phi_2}\left(\frac{n^{10} Z_n(\phi_2)}{w_{\min}^2 \gamma^2}\right)$. We note that the values $f^{(1)}(i \to j), f^{(2)}(i \to j)$ in the following Lemma are defined with respect to $\widehat{w}_1 \mathcal{M}\left(\widehat{\phi}_1, \pi_1\right) \oplus \widehat{w}_2 \widehat{\mathcal{M}}\left(\widehat{\phi}_2, \pi_2\right)$.

**Lemma 10.2.** *Given access to an oracle for $\widehat{\mathcal{M}}$ and rankings $\widehat{\pi}_1$ and $\widehat{\pi}_2$ which agree with $\pi_1$ and $\pi_2$ in the first $r_1$ and $r_2$ elements respectively, where $r_1 \geq \log_{1/\phi_1}\left(\frac{n^{10} Z_n(\phi_1)}{w_{min}^2 \gamma^2}\right)$ and $r_2 \geq \log_{1/\phi_2}\left(\frac{n^{10} Z_n(\phi_2)}{w_{min}^2 \gamma^2}\right)$, then procedure RECOVER-REST with $\epsilon = \frac{1}{10}\gamma$, outputs the rankings $\pi_1$ and $\pi_2$ with high probability.*

*Proof.* First suppose that the condition in Step 2 of Recover-Rest is true for some $e_{i^*}$. This would imply that $f^{(2)}(i^* \to 1) < \frac{\widehat{w}_1}{\widehat{w}_2} \frac{f^{(1)}(i^* \to 1)}{n^2 g(n, \widehat{\phi}_1)}$. Hence, conditioned on $e_{i^*}$ going to the first position, the new weight $w'_1$ would be $\frac{1}{1 + \frac{\widehat{w}_2}{\widehat{w}_1} \frac{f^{(2)}(i^* \to 1)}{f^{(1)}(i^* \to 1)}} \geq 1 - \frac{1}{ng(n, \widehat{\phi}_1)}$. Since, $g(n, \widehat{\phi}_1)$ is an upper bound on the sample complexity of learning a single Mallows model with parameter $\widehat{\phi}_1$, with high probability we will only see samples from $\pi_1$ and from the guarantees of Lemma 10.7 and Lemma 10.8, we will recover both the permutations. A similar analysis is also true for step 4 of RECOVER-REST. If none of the above conditions happen, then step 5 will succeed because of the guarantee from Lemma 3.2.

Next we will argue about the correctness of the linear equations in step 6. We have set a threshold $\delta = \frac{w_{\min} \gamma^2}{n^4}$, from Lemma 10.1, we know that the linear equations are correct up to error $\delta$. Once we have obtained good estimates for $f^{(1)}(i \to j)$ for all $e_i$ and $j > r$, Lemma 10.3 implies that

step 7 of RECOVER-REST will give us the correct ranking $\tau_1$. This combined with Lemma 10.8 will recover both the rankings with high probability. $\qquad\square$

We now present the Lemmas needed in the proof of the previous Lemma 10.2.

**Lemma 10.3.** *Consider a length $n$ Mallows model with parameter $\phi$. Consider an element $e_i$ and let $pos(e_i) = j$. Let $f(i \to k) = Pr[e_i \mapsto k]$. Then we have*

1. *$f(i \to k)$ is maximum at $k = j$.*

2. *For all $k > j$, $f(i \to k - 1) \geq f(i \to k)(1 + gain(\phi))$.*

3. *For all $k < j$, $f(i \to k) \geq f(i \to k - 1)(1 + gain(\phi))$.*

*Here $gain(\phi) = \frac{(1-\phi)}{4\phi} min(\frac{1}{n}, 1 - \phi^2)$.*

*Proof.* The case $j = 1$ is easy. Let $j > 1$ and consider the case $k > j$. Let $S_k = \{\pi : pos_\pi(e_i) = k\}$. Similarly let $S_{k-1} = \{\pi : pos_\pi(e_i) = k - 1\}$. For a set $U$ of rankings, let $p(U) = Pr[\pi \in U]$. Notice that $f(i \to k - 1) = p(S_{k-1})$ and $f(i \to k) = p(S_k)$. Let $X = \{e_j : pos_{\pi^*}(e_j) > pos_{\pi^*}(e_i)\}$ and $Y = \{e_j : pos_{\pi^*}(e_j) < pos_{\pi^*}(e_i)\}$. We will divide $S_k$ into 4 subsets depending on the elements $\tau_1$ and $\tau_2$ which appear in positions $(k-1)$ and $(k-2)$ respectively. In each case we will also present a bijection to the rankings in $S_{k-1}$.

- $S_{k,1} = \{\pi \in S_k : \tau_1, \tau_2 \in X\}$. For each such ranking in $S_k$ we form a ranking in $S_{k-1}$ by swapping $e_i$ and $\tau_1$. Call the corresponding subset of $S_{k-1}$ as $S_{k-1,1}$.

- $S_{k,2} = \{\pi \in S_k : \tau_1 \in X, \tau_2 \in Y\}$. For each such ranking in $S_k$ we form a ranking in $S_{k-1}$ by swapping $e_i$ and $\tau_1$. Call the corresponding subset of $S_{k-1}$ as $S_{k-1,2}$.

- $S_{k,3} = \{\pi \in S_k : \tau_1 \in Y, \tau_2 \in X\}$. For each such ranking in $S_k$ we form a ranking in $S_{k-1}$ by swapping $e_i$ and $\tau_1$. Call the corresponding subset of $S_{k-1}$ as $S_{k-1,3}$.

- $S_{k,4} = \{\pi \in S_k : \tau_1, \tau_2 \in Y\}$. Consider a particular ranking $\pi$ in $S_{k,4}$. Notice that since $e_i$ is not in it's intended position there must exist at least one element $x \in X$ such that $pos_\pi(x) < pos_\pi(e_i)$ in $S_k$. Let $x^*$ be such an element with the largest value of $pos_\pi(x)$. Let $y \in Y$ be the element in the position $pos_\pi(x^*) + 1$. For each such ranking in $S_k$ we form a ranking in $S_{k-1}$ by swapping $e_i$ and $\tau_1$ and $x^*$ and $y$. Call the corresponding subset of $S_{k-1}$ as $S_{k-1,4}$.

It is easy to see that the above construction gives a bijection from $S_k$ to $S_{k-1}$. We also have the following

- $p(S_{k-1,1}) = \frac{1}{\phi} p(S_{k,1})$. This is because the swap is decreasing the number of inversions by exactly 1.

- $p(S_{k-1,2}) = \frac{1}{\phi} p(S_{k,2})$. This is because the swap is decreasing the number of inversions by exactly 1. $p(S_{k-1,3}) = \phi p(S_{k,3})$. This is because the swap is increasing the number of inversions by exactly 1. $p(S_{k-1,4}) = p(S_{k,4})$. This is because the two swaps maintain the number of inversions.

Also note that there is a bijection between $S_{k,2}$ and $S_{k,3}$ such that every ranking in $S_{k,3}$ has one more inversion than the corresponding ranking in $S_{k,2}$. Hence we have $p(S_{k,3}) = \phi p(S_{k,2})$.

Now we have

$$f(i \to k-1) = \sum_i p(S_{k-1,i}) \tag{7}$$

$$= \frac{1}{\phi}p(S_{k,1}) + \frac{1}{\phi}p(S_{k,2}) + \phi p(S_{k,3}) + p(S_{k,4}) \tag{8}$$

$$= f(i \to k) + p(S_{k,1})(\frac{1}{\phi} - 1) + p(S_{k,2})(\frac{1}{\phi} - 1) - p(S_{k,3})(1 - \phi) \tag{9}$$

$$\tag{10}$$

If $p(S_{k,1}) \geq \frac{1}{4}p(S_k)$ or $p(S_{k,2}) \geq \frac{1}{4}p(S_k)$, then $gain(\phi) \geq \frac{(1-\phi)}{4\phi}(1 - \phi^2)$. If not, then we have $p(S_{k,4}) \geq 1/4$. Divide $S_{k,4}$ as $\cup_j S_{k,4,j}$ where $S_{k,4,j} = \{\pi \in S_{k,4} : pos_\pi(x^*) = j\}$. It is easy to see that $p(S_{k,4,j}) = \phi(S_{k,4,j-1})$. Hence we have $p(S_{k,2}) > p(S_{k,3}) > \frac{1}{n}p(S_{k,4}) \geq \frac{1}{4n}$. In this case we will have $gain(\phi) \geq \frac{(1-\phi)}{4n\phi}(1 - \phi^2)$.

The case $k < j$ is symmetric. $\qquad\square$

**Lemma 10.4.** *Consider a length $n$ Mallows model with parameter $\phi$. Let the target ranking be $\pi^* = (e_1, e_2, \ldots, e_n)$. Let $f^{(1)}(i \to j)$ be the probability that the element at position $i$ goes to position $j$. We have for all $i, j$*

$$f^{(1)}(i \to j) = f^{(1)}(j \to i)$$

*Proof.* We will prove the statement by induction on $n$. For $n = 1, 2$, the statement is true for all $\phi$. Now assume it is true for all $n \leq l - 1$. Consider a length $l$ Mallows model. We have

$$f^{(1)}(i \to j) = \sum_{k \leq i} f^{(1)}(i - 1 \to j - 1|e_k \to 1) Pr(e_k \mapsto 1) + \sum_{j \geq k > i} f^{(1)}(i \to j - 1|e_k \to 1) Pr(e_k \mapsto 1)$$

$$+ \sum_{k > j} f^{(1)}(i \to j|e_k \to 1) Pr(e_k \mapsto 1)$$

$$= \sum_{k \leq i} f^{(1)}(j - 1 \to i - 1|e_k \to 1) Pr(e_k \mapsto 1) + \sum_{j \geq k > i} f^{(1)}(j - 1 \to i|e_k \to 1) Pr(e_k \mapsto 1)$$

$$+ \sum_{k > j} f^{(1)}(j \to i|e_k \to 1) Pr(e_k \mapsto 1)$$

$$= f^{(1)}(j \to i)$$

$\qquad\square$

**Lemma 10.5.** *Consider a length $n$ Mallows model with parameter $\phi$. Let the target ranking be $\pi^* = (e_1, e_2, \ldots, e_n)$. Consider a position $i$ which has element $e_i$.*

1. *$f(j \to i)$ is maximum at $j = i$.*

2. *For all $k > i$, $f(k - 1 \to i) \geq f(k \to i)(1 + gain(\phi))$.*

3. *For all $k < i$, $f(k \to i) \geq f(k - 1 \to i)(1 + gain(\phi))$.*

*Here $gain(\phi) = \frac{(1-\phi)}{4\phi}min(\frac{1}{n}, 1 - \phi^2)$.*

*Proof.* Follows from Lemmas 10.3 and 10.4. $\qquad\square$

**Lemma 10.6.** *Given access to $m = O(\frac{1}{gain(\phi)^2}\log(\frac{n}{\delta}))$ samples $w_1\mathcal{M}(\phi_1, \pi_1) \oplus w_2\mathcal{M}(\phi_2, \pi_2)$, with $\phi_1 = \phi_2$, procedure* REMOVE-COMMON-PREFIX *with $\epsilon = \frac{1}{10}gain(\phi)$, succeeds with probability $1 - \delta$.*

*Proof.* If the two permutations have the same first element $e_1$, then we have $x_1 = 1/Z_n(\phi)$. Since $m$ is large enough, all our estimates will be correct up to multiplicative error of $\sqrt{1 + gain(\phi)}$. By induction, assume that the two permutations have the same prefix till $t - 1$. By the property of

the Mallows model, we know that the remaining permutations are also a mixture of two Mallows models with the same weight. Hence, at step $t$, if we estimate each probability within multiplicative factor of $\sqrt{1 + gain(\phi)}$, we will succeed with high probability. $\qquad\square$

**Lemma 10.7.** *Given access to $m = O(\frac{1}{gain(\phi)^2} \log(\frac{n}{\delta}))$ samples from a Mallows model $\mathcal{M}(\phi, \pi)$, procedure* LEARN-SINGLE-MALLOW *with $\epsilon = \frac{1}{10} gain(\phi)$, succeeds with probability $1 - \delta$.*

*Proof.* In order to learn, it is enough to estimate $f(i \to j) = Pr[e_i$ goes to position $j]$ for every element $e_i$ and position $j$. Having done that we can simply assign $\text{pos}(e_i) = \arg\max_j f(i \to j)$. From Lemma 10.3 we know that this probability is maximum at the true location of $e_i$ and hence is at least $1/n$. Hence, it is enough to estimate all $f(i \to j)$ which are larger than $1/n$ up to multiplicative error of $\sqrt{1 + gain(\phi)}$. By standard Chernoff bounds, it is enough to sample $O(\frac{1}{gain(\phi)^2} \log(\frac{n}{\delta}))$ from the oracle for $\mathcal{M}(\phi, \pi)$. $\qquad\square$

**Lemma 10.8.** *Given the parameters of a mixture model $w_1 \mathcal{M}(\phi_1, \pi_1) \oplus w_2 \mathcal{M}(\phi_2, \pi_2)$ and one of the permutations $\pi_1$, procedure* Find-Pi *with $\epsilon = \frac{w_{min}\gamma}{10}$, succeeds with probability $1 - \delta$. Here $\gamma = \min(gain(\phi_1), gain(\phi_2))$.*

*Proof.* For any element $e_i$ and position $j$, we have that

$$f(i \to j) = w_1 f^{(1)}(i \to j) + w_2 f^{(2)}(i \to j). \tag{11}$$

Here $f^{(1)}(i \to j)$ is the probability that element $e_i$ goes to position $j$ in $\mathcal{M}(\phi_1, \pi_1)$. Similarly, $f^{(2)}(i \to j)$ is the probability that element $e_i$ goes to position $j$ in $\mathcal{M}(\phi_2, \pi_2)$. We can compute $f^{(1)}(i \to j) = f^{(1)}_{(n,j,i)}$ using dynamic programming via the following relation

$$f^{(1)}_{(n,1,i)} = \phi_1^{i-1}/Z_n(\phi_1)$$

$$f^{(1)}_{(n,l,i)} = \frac{1}{Z_n(\phi_1)} \left[ \left( \sum_{j=1}^{i-1} \phi_1^{j-1} \right) f^{(1)}_{(n-1,l-1,i-1)} + \left( \sum_{j=i+1}^{n} \phi_1^{j-1} \right) f^{(1)}_{(n-1,l-1,1)} \right]$$

Here $f^{(1)}_{(n,l,i)}$ is the probability that the element at the $i$th position goes to position $l$ in a length $n$ Mallows model. Notice that this probability is independent of the underlying permutation $\pi$. Having computed $f^{(1)}(i \to j)$ using the above formula, we can solve Equation 11 to get $f^{(2)}(i \to j)$ to accuracy $\sqrt{1 + w_{min}\gamma}$ and figure out $\pi_2$. The total number of samples required will be $O(\frac{1}{\gamma^2 w_{min}^2} \log(\frac{n}{\delta}))$. $\qquad\square$

## 11 Wrapping up the Proof

*Proof of Theorem 3.1.* Let $\epsilon_s$ be the entry-wise error in $P$ from the estimates. From Lemma 13.3, $\epsilon_s < 3 \log n/\sqrt{N}$. We aim to estimate each of the parameters $\phi_1, \phi_2 w_1, w_2$ up to error at most $\epsilon$. Let for convenience, $\gamma = \frac{(1-\phi_{max})^2}{4n\phi_{max}}$.

Let $\epsilon = \min \{\epsilon, \epsilon_0\}$. Let $\epsilon_3 = \vartheta_{9.9}(n, \phi_{min}, \epsilon)$. Let us also set $\epsilon_2 = \vartheta_{9.1}(n, \epsilon_3, \phi_{min}, w_{min})$. Let $\epsilon_2'$ be a parameter chosen large enough such that $\epsilon_2' \geq \frac{\epsilon_2}{\gamma_{min}} + \vartheta_{8.1}$, and $\epsilon \leq \vartheta_{3.2}(n, \epsilon_2', \epsilon_s, w_{min}, \phi_{min})$.

In the non-degenerate case, suppose there is a partition such that $\sigma_2(M_a), \sigma_2(M_b), \sigma_2(M_c) \geq \epsilon_2'$, Lemma 8.1 guarantees that $\sigma_2(M_a'), \sigma_2(M_b'), \sigma_2(M_c') \geq \epsilon_2$. In this case, Lemma 3.2 ensures that one of the $O(\log n)$ rounds of the algorithm succeeds and we get the parameters $w_1, w_2, \phi_1, \phi_2$ within an error $\epsilon$ using Lemma 3.2. Further, Lemma 3.2 will also find the top $r, s$ elements of $\pi_1$ and $\pi_2$ respectively where $r = \log_{1/\phi_1} \left( \frac{n^{10}}{\gamma^2 w_{min}} \right)$ and $s = \log_{1/\phi_2} \left( \frac{n^{10}}{\gamma^2 w_{min}} \right)$. We will then appeal to Lemma 10.2 (along with Lemma 2.6) to recover the entire rankings $\pi_1, \pi_2$.

Lemma 2.6 implies that the total variation distance between distributions of $w_1 \mathcal{M}(\phi_1, \pi_1) \oplus w_2 \mathcal{M}(\phi_2, \pi_2)$ and $\widehat{w}_1 \mathcal{M}\left(\widehat{\phi}_1, \pi_1\right) \oplus \widehat{w}_2 \widehat{\mathcal{M}}\left(\widehat{\phi}_2, \pi_2\right)$ is at most $\frac{\epsilon n^2}{\phi_{min}}$. Since $\epsilon \leq \epsilon_0$, this variation distance is at most $\frac{\phi_{min}}{10n^3 S(w_{min}/2, \sqrt{\phi_{max}})}$. Here $S(w_{min}/2, \sqrt{\phi_{max}})$ is an upper bound on the

number of samples needed by RECOVER-REST to work given true parameters (and not estimations). This allows to analyze the performance of RECOVER-REST assuming that we get perfect estimates of the parameters $(w_1, w_2, \phi_1, \phi_2)$ since samples used by RECOVER-REST which are drawn from $w_1 \mathcal{M}(\phi_1, \pi_1) \oplus w_2 \mathcal{M}(\phi_2, \pi_2)$ will be indistinguishable from samples from $\widehat{w}_1 \mathcal{M}\left(\widehat{\phi}_1, \pi_1\right) \oplus \widehat{w}_2 \widehat{\mathcal{M}}\left(\widehat{\phi}_2, \pi_2\right)$ except with probability $\frac{1}{10n^3}$. This followed by the guarantee of Lemma 10.2 will recover the complete rankings $\pi_1$ and $\pi_2$.

In the degenerate case, due to our choice of $\epsilon_2$, Lemma 9.1 shows that $\widehat{\phi}$ is $\epsilon_3$ close to both $\phi_1$ and $\phi_2$. Using Lemma 9.9 we then conclude that step 4 of Algorithm 1 recovers $\pi_1, \pi_2$ and the parameters $w_1, w_2$ within error $\epsilon$. $\qquad \square$

## 12  Conclusions and Future Directions

In this paper we gave the first polynomial time algorithm for learning the parameters of a mixture of two Mallows models. Our algorithm works for an arbitrary mixture and does not need separation among the underlying base rankings. We would like to point out that we can obtain substantial speed-up in the first stage (tensor decompositions) of our algorithm by reducing to an instance with just $k \sim \log_{1/\phi} n$ elements.

Several interesting directions come out of this work. A natural next step is to generalize our results to learn a mixture of $k$ Mallows models for $k > 2$. We believe that most of these techniques can be extended to design algorithms that take $\text{poly}(n, 1/\epsilon)^k$ time. It would also be interesting to get algorithms for learning a mixture of $k$ Mallows models which run in time $\text{poly}(k, n)$, perhaps in an appropriate smoothed analysis setting [23] or under other non-degeneracy assumptions. Perhaps, more importantly, our result indicates that tensor based methods which have been very popular for problems such as mixture of Gaussians, might be a powerful tool for solving learning problems over rankings as well. We would like to understand the effectiveness of such tools by applying them to other popular ranking models as well.

## 13  Some Useful Lemmas for Error Analysis

**Lemma 13.1.** *Let $u, u', v, v'$ denote vectors and fix parameters $\delta, \gamma > 0$. Suppose $\|u \otimes v - u' \otimes v'\|_F < \delta$, and $\gamma \le \|u\|, \|v\|, \|u'\|, \|v'\| \le 1$,*
*with $\delta < \frac{\gamma^2}{2}$. Given a decomposition $u = \alpha_1 u' + u^\perp$ and $v = \alpha_2 v' + v^\perp$, where $u^\perp$ and $v^\perp$ are orthogonal to $u', v'$ respectively, then we have*

$$\|u^\perp\| < \sqrt{\delta} \text{ and } \|v^\perp\| < \sqrt{\delta}.$$

*Proof.* We are given that $u = \alpha_1 u' + u^\perp$ and $v = \alpha_2 v' + v^\perp$. Now, since the tensored vectors are close

$$\|u \otimes v - u' \otimes v'\|_F^2 < \delta^2$$
$$\|(1 - \alpha_1\alpha_2)u' \otimes v' + \alpha_2 u^\perp \otimes v' + \alpha_1 u' \otimes v^\perp + u^\perp \otimes v^\perp\|_F^2 < \delta^2$$
$$\gamma^4(1 - \alpha_1\alpha_2)^2 + \|u^\perp\|^2 \alpha_2^2 \gamma_{\min}^2 + \|v^\perp\|^2 \alpha_1^2 \gamma^2 + \|u^\perp\|^2 \|v^\perp\|^2 < \delta^2 \qquad (12)$$

This implies that $|1 - \alpha_1\alpha_2| < \delta/\gamma^2$.

Now, let us assume $\beta_1 = \|u^\perp\| > \sqrt{\delta}$. This at once implies that $\beta_2 = \|v^\perp\| < \sqrt{\delta}$. Hence one of the two (say $\beta_2$) is smaller than $\sqrt{\delta}$.
Also

$$\gamma^2 \le \|v\|^2 = \alpha_2^2 \|v'\|^2 + \beta_2^2$$
$$\gamma^2 - \delta \le \alpha_2^2$$
$$\text{Hence,} \quad \alpha_2 \ge \frac{\gamma}{2}$$

Now, using (12), we see that $\beta_1 < \sqrt{\delta}$. $\qquad \square$

**Lemma 13.2.** *Let $\phi \in (0,1)$ be a parameter and denote $c_2(\phi) = \frac{Z_n(\phi)}{Z_{n-1}(\phi)} \frac{1+\phi}{\phi}$ and $c_3(\phi) = \frac{Z_n^2(\phi)}{Z_{n-1}(\phi)Z_{n-2}(\phi)} \frac{1+2\phi+2\phi^2+\phi^3}{\phi^3}$. Then we have that $1 \le c_2(\phi) \le 3/\phi$ and $1 \le c_3(\phi) \le 50/\phi^3$.*

*Proof.* Since $0 < \phi < 1$, we have that $Z_{n-1}(\phi) \le \frac{1}{1-\phi}$. Observe that $1 \le \frac{Z_n(\phi)}{Z_{n-1}(\phi)} \le 1 + \frac{1}{Z_{n-1}(\phi)} \le 2$. The bounds now follow immediately. $\square$

**Lemma 13.3.** *In the notation of section 2, given $N$ independent samples, the empirical average $\widehat{P}$ satisfied $\|P - \widehat{P}\|_\infty < \sqrt{C\frac{\log n}{N}}$ with probability $1 - n^{-C/8}$.*

*Proof.* This follows from a standard application of Bernstein inequality followed by a union bound over the $O(n^3)$ events. $\square$