[Reviews · NeurIPS 2014]

Submitted by Assigned_Reviewer_20

The paper gives an algorithm to learn the parameters of a mixture of 2 Mallows models, which describes a distribution over permutations. This also proves the identifiability of the model. The algorithm is a spectral/ method of moments type algorithm, and ensures that the time and samples required is polynomial in the number of objects to be ranked.

Overall, the paper is well written and gives a good intuitive overview of the results and proof techniques. The results look very plausible, and the intuition provided is believable, however I have not gone through the proofs in the Appendix.

Some minor comments:

Line 283: Algorithm 1: The matrix $M_a'$ seems undefined. I am guessing it is the 2 column $[u,v]$ matrix, it might be good to specify it.

Line 303: up to $\epsilon$ accuracy. -- The meaning of this is not clear -- in an earlier statement it was mentioned that the mixing proportions, and the phi parameters were individually learnt to $\epsilon$ accuracy and the permutations are learnt exactly. Is this what the statement means? If so it might be better to explicitly mention it.

Line 330: The hats on $w_i% and $\phi_i$ are a bit awkward, it might be better to use $\hat w_1$ instead of $\hat{w_1}$

Line 338: The algorithm uses the vectors $\hat x$ and $\hat y$, which are not among the input arguments.

Line 410: Not any integral solution to the equation will do -- you need to restrict $x_1$ to be less than $n$, and $x_2$ less than $n-1$ and so on.

Summary: Reasonably well written paper on learning the parameters to a mixture of Mallows model using a spectral type algorithm. Would make a nice addition to the conference.

Submitted by Assigned_Reviewer_21

Summary:
The authors consider the problem of estimating the parameters of a mixture of two Mallows models. Finding the maximum likelihood estimation of the mixture models in polynomial time has been an open problem. The authors solve the open problem and presents the first polynomial time algorithms for estimating MLE of a mixture of two Mallows models. Further, the authors validates effectiveness of the proposed algorithm by comparing it with a standard EM algorithm in some experiments.

Comments:

The problem considered in the paper is quite relevant to the community. The technical contribution of the paper is non-trivial and looks interesting.The paper is well organized and well written.

I am curious about the relationship between the techniques paper and those used for learning the mixture of constant number of Gaussian distributions. Intuitively, both two problems share a common structure: learning mixture coefficients and estimating "center" points. Discussing the relationship might be useful to understand the technical contribution of this paper deeper.

Summary: I think the paper is interesting enough and deserves an acceptance.

Submitted by Assigned_Reviewer_49

This paper considers the problem of the estimation of a mixture of two Mallows models. Though this problem has already attracted significant attention in the literature, the present paper is the first to introduce a polynomial time algorithm that provably solve it. It shows in particular the identifiability of the model, which was not known previously. The algorithm relies on two main ingredients:
• Explicit combinatorial formulas for the probability of the set of the k items ranked first under the mixture, for k = 1, 2, 3.
• The tensor decomposition of the matrix of these probabilities in terms of the representative vectors of the two Mallows models of the mixture.
The statement of the algorithm is rather technical but it is well segmented into subroutines. Broadly speaking, the algorithm first randomly partition the set of items to obtain the mixture parameters as well as the diffusion parameters and the first elements of the central rankings of the two Mallows models, via a tensor decomposition. Then it completes the central permutations via a fine analysis of the different possible cases. The algorithm is proved to find the exact parameters of the mixture with high probability. At last, numerical experiments on synthetic data show that the proposed algorithm largely outperforms a classic EM based algorithm of the literature.

This paper introduces explicit calculations on the Mallows model that, up to our knowledge, were never carried out before, highlighting a particular tensor-based structure. They are then exploited with a fine technical analysis in the design of a specific, novel algorithm. Finally the technical complexity of the approach is rewarded by convincing numerical experiments.
Summary: The novelty of the approach and the deepness of the calculations promise a significant impact to the results of this paper.
Author Feedback
Author rebuttal: We thank all the reviewers for their thoughtful reviews.

Reviewer 1: Thanks for your suggestions on the writeup. We will incorporate them in the final version. (Regarding line 303: yes,
that's right. We'll make the text clearer.)

Reviewer 2: Thanks, we will add a discussion. In the case of mixtures of Gaussians, there is a natural notion of moments of the Gaussian which have a very nice algebraic form in terms of the parameters of the model, which are exploited to design algorithms (the Method of Moments). For the Mallows model which deals with discrete objects like permutations, there is no such natural notion of 2nd/3rd/i-th order moment. One contribution of our work is come up with the right statistics of the model (Pr[elements {e_1,e_2,e_3} is ranked in positions {1,2,3}]) that have closed form expressions with a nice algebraic structure -- we leverage to obtain good preliminary estimates for the model parameters. We will clarify this comparison better in the final version.

Reviewer 3: Thank you for your thoughtful review.